# SIPDO: Closed-Loop Prompt Optimization via Synthetic Data Feedback

**Yaoning Yu**[1]* **Ye Yu**[1]* **Peiyan Zhang**[2] **Kai Wei**[3] **Haojing Luo**[2] **Haohan Wang**[1]
[1]University of Illinois at Urbana-Champaign, [2]Starc.Institute, [3]University of South Florida

## Abstract

Prompt quality plays a critical role in the performance of large language models (LLMs), motivating a growing body of work on prompt optimization. Most existing methods optimize prompts over a fixed dataset, assuming static input distributions and offering limited support for iterative improvement. We introduce **SIPDO** (**S**elf-**I**mproving **P**rompts through **D**ata-Augmented **O**ptimization), a closed-loop framework for prompt learning that integrates synthetic data generation into the optimization process. SIPDO couples a synthetic data generator with a prompt optimizer, where the generator produces new examples that reveal current prompt weaknesses and the optimizer incrementally refines the prompt in response. This feedback-driven loop enables systematic improvement of prompt performance without assuming access to external supervision or new tasks. Experiments across question answering and reasoning benchmarks show that SIPDO outperforms standard prompt tuning methods, highlighting the value of integrating data synthesis into prompt learning workflows.

## 1 Introduction

Large language models (LLMs) have demonstrated strong performance across a wide range of natural language tasks, including classification, question answering, and reasoning Liu et al. (2025); Chen et al. (2025a). However, their output quality is highly sensitive to prompt design—small changes in phrasing, structure, or formatting can lead to significant variations in performance (He et al., 2024; Spiess et al., 2025). This sensitivity has made prompt optimization a core challenge in adapting LLMs to downstream applications, where consistency and reliability are crucial. In domains such as healthcare and finance, the inability to ensure stable, predictable performance makes prompt optimization more than a desirable enhancement, but a critical necessity for reliable system deployment.

The core challenge of prompt optimization is multifaceted. Unlike traditional hyperparameter tuning, the search space of prompts is discrete, non-differentiable, and vast. Small modifications to prompts can have unpredictable effects on LLMs' behavior, and the gradient-based methods that typically power optimization in machine learning are not directly applicable. Furthermore, LLMs often perform well on a fixed, curated test set, but their performance can deteriorate when faced with novel linguistic variations, edge cases, or adversarial queries. This makes prompt optimization particularly challenging, as a prompt that performs well in one scenario may fail when the input distribution shifts, leading to issues such as catastrophic forgetting or fragile performance across different contexts.

Prior work in prompt optimization has explored manual tuning, discrete search, and gradient-based methods to improve model responses (Wang et al., 2023; Shin et al., 2020; Cui et al., 2024; Kwon et al., 2024; Zhang et al., 2024; Chen et al., 2025b). While effective in some settings, these approaches do not address the dynamic nature of real-world inputs, where the input space evolves over time. As a result, they can produce prompts that perform well on average but lack robustness when the input distribution changes.

In contrast, data augmentation has long been used in supervised learning to improve robustness by exposing models to diverse conditions (Mikołajczyk & Grochowski, 2018; Wang et al., 2022). In

---

*These authors contributed equally.

the context of prompt learning, the ability of LLMs to generate high-quality synthetic data presents an exciting opportunity to improve prompt optimization. However, existing prompt optimization methods do not leverage synthetic data in a dynamic, feedback-driven manner (Singh et al., 2023; Gilardi et al., 2023; Tang et al., 2023; Gao et al., 2023; Yu et al., 2025). Moreover, it is not sufficient to simply produce more data; the challenge is to generate data that is purposeful and stressful that targets the current failure modes of the prompt and provides a progressive challenge that helps guide its evolution. The synthetic data must also be carefully crafted to ensure that it does not overwhelm the model with trivially easy or overly difficult cases, but instead exposes latent weaknesses that need to be addressed.

To address these challenges, we propose **SIPDO** (**S**elf-**I**mproving **P**rompts through **D**ata-Augmented **O**ptimization), a closed-loop framework for prompt optimization that integrates synthetic data generation directly into the learning process. SIPDO consists of two components: a *Synthetic Data Generator* that produces inputs specifically designed to challenge the current prompt, and a *Prompt Optimizer* that uses these examples to iteratively refine the prompt. This feedback loop enables the prompt to evolve continuously over time, adapting to new, previously unseen inputs with minimal supervision for each new scenario individually by providing a general idea of the task type and what needs to be in mind without manually constructing a new prompt. Particularly, **SIPDO** addresses the unique challenges of prompt optimization by transforming the optimization process from a static, one-time procedure to a dynamic, self-adaptive learning loop. This shift is critical for ensuring prompt robustness in the face of evolving input distributions.

**Contributions.** This paper makes the following contributions:

- We introduce a feedback-driven framework SIPDO that integrates synthetic data generation into prompt optimization, providing a novel pathway for improving prompt robustness.
- We develop a method to construct synthetic examples that dynamically stress-test prompts, revealing failure modes and guiding refinement.
- We empirically demonstrate that augmenting prompt optimization with synthetic data improves performance across multiple reasoning benchmarks, surpassing existing prompt tuning methods.

## 2 RELATED WORK

### 2.1 AUTOMATIC PROMPT ENGINEERING

Automatically discovering optimal prompts has become a key challenge in the era of large language models (LLMs). Automatic Prompt Engineering (APE) employs optimization-based, generative, and template-driven approaches. Optimization techniques include gradient-based search (Shin et al., 2020), reinforcement learning (Ouyang et al., 2022; Kwon et al., 2024), and evolutionary algorithms (Cui et al., 2024). Generative methods use models like GPT and Gemini to generate candidate prompts, with StablePrompt (Kwon et al., 2024) optimizing prompts via reinforcement learning. Additionally, PromptAgent (Wang et al., 2023) breaks down prompt creation into sub-goals, while template-driven approaches, like fill-in-the-blank formats, ensure clarity (Chen et al., 2024). Recent work has expanded on automatic prompt optimization techniques. REVOLVE (Zhang et al., 2024) treats prompt optimization as an iterative text editing process that uses changes in model outputs across rounds to make updates more stable. In addition, Chen et al. (2025b) stresses that good prompts should give consistent results despite randomness, and proposes measuring semantic stability to guide prompt improvement. In parallel, AutoPDL (Spiess et al., 2025) automates the discovery of optimal configurations for agents which successive halving to explore the space of agentic and non-agentic prompting patterns. The sequential optimal learning approach for automated prompt engineering (Wang et al., 2025) uses Bayesian regression and Knowledge-Gradient policies to efficiently identify effective prompt features. Progressively Automatic Prompt Optimization (Qu et al., 2025) introduces an evolution-based algorithm to optimize prompts for visual classification tasks.

Adopting LLM as a feedback loop to refine prompts has recently emerged. Self-Refine (Madaan et al., 2023) improves the model by generating feedback based on the previously generated outputs and using that feedback to refine the next output. In Promptbreeder (Fernando et al., 2023), it uses an iterative feedback loop to select better prompts from the original prompts and mutation prompts, while Recursive In-Context Learning for Autonomous Prompt Generation in Large Language Models: A Self-Instructed Approach (Yilar et al., 2024) introduces a framework that refines prompts

through iterative loops based on generated outputs. More recently, SPO(Self-Supervised Prompt Optimization) (Xiang et al., 2025) uses iterative feedback to refine prompts by comparing the outputs of the current prompt and its revised version without relying on ground truth, and DLPO (Peng et al., 2025) optimizes prompts within loops by mimicking a deep learning style where it uses textual loss, gradients, and a feedback process to update prompts. CriSPO (He et al., 2025) also extends iterative feedback loop optimization using critique guides to update prompts with multi-metric.

We propose a hybrid framework integrating LLM-driven rewriting with natural language feedback (Pryzant et al., 2023), alongside self-reflection (Shinn et al., 2024) and planning (Wang et al., 2023), enhancing prompt adaptability and precision.

## 2.2 DATA SYNTHESIS

Using large language models (LLMs) for data synthesis is a relatively new and rapidly evolving approach. Recent advancements have shown that LLMs possess the capability to generate text with fluency and quality comparable to human output (Li et al., 2023; Mukherjee et al., 2023; Eldan & Li, 2023). For instance, prior work (Gao et al., 2023) has explored leveraging pre-trained language models (PLMs) to generate task-specific text data that can be used to train and evaluate. Recent work Magpie (Xu et al., 2024) leverages the auto-regressive nature of aligned LLMs to generate high-quality instruction data. Additionally, Synthetic Text Generation for Training Large Language Models via Gradient Matching (Nguyen et al., 2025) proposes a novel approach to generate synthetic text that matches the gradients of human data. However, these studies have not fully incorporated advanced methodologies such as chain-of-thought (CoT) reasoning, in-context learning, or data synthesis driven by prompts that integrate task descriptions and label information.

In this study, we systematically experimented with a range of techniques, including in-context learning and prompt-driven data synthesis, combining task descriptions and label information. Our results show that these approaches generate high-quality synthetic data. By introducing a difficulty tier, we further enhanced the data's robustness and applicability. These findings demonstrate the potential of combining advanced LLM capabilities with tailored prompting strategies to improve data synthesis for prompt optimization.

## 3 METHOD

In this work, we introduce SIPDO, a two-agent system for optimizing prompts using data augmentation techniques. The workflow has two cooperating agents: (i) Data Generator creates synthetic data with increasing difficulty levels to expose weaknesses in the prompt, and (ii) Auto Prompt Optimizer iteratively analyzes errors and rewrites the prompt to maximize task performance. An overview of SIPDO is shown in Fig 1.

**Notation.** We define the true data distribution as $S$, which governs input-label pairs $(x, y) \in \mathcal{X} \times \mathcal{Y}$. Let $N$ denote the size of an i.i.d. dataset drawn from $S$, denoted as $\{(x_i, y_i)\}_{i=1}^{N} \sim S$. We consider LLMs equipped with a prompt $p \in \mathcal{P}$, and define its output function as $f(p, x) \in \mathcal{Y}$. Prediction accuracy is measured using a bounded surrogate loss $L\big(f(p, x), y\big)$, where $L \in [0, 1]$. We introduce a synthetic data generator defined by a distribution $q_\psi(\tilde{x}, \tilde{y})$, parameterized by $\psi \in \Psi$, which produces synthetic samples forming a dataset $D = \{(\tilde{x}_i, \tilde{y}_i)\}_{i=1}^{M}$, where $M$ is the number of generated examples. To ensure that the synthetic labels remain realistic, we estimate the population label prior with $p^*(y)$ and use this to regularize the generator.

## 3.1 DATA GENERATOR

The Data Generator supplies fresh, well-targeted examples that expose the weakness by creating a new synthetic-pair whose difficulty is designed beyond prompt's current reach.

**Sampling rule.** The data generator first draws a target label $\tilde{y} \sim p^*(y)$. By sampling a latent variable $z \sim g_\phi(z|S)$ that captures the structure of few-shot $S$, the decoder $q_\psi$ produces $\tilde{x} = q_\psi(z, \tilde{y}, c)$ where c is a controlled difficulty tier.

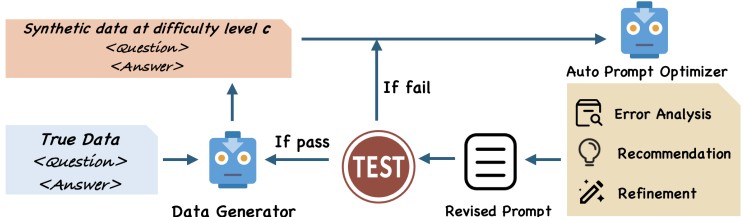

Figure 1: Starting from true data distribution $S$, the Data Generator(left) produces a synthetic question-answer pair at difficulty level $c$. The Auto Prompt Optimizer(right) evaluates the current prompt on this synthetic data via three sub-modules-error analysis, recommendation, and refinement-and outputs a revised prompt. The revised prompt is tested on present failures and all previously solved examples. If the prompt still makes errors, then return to the Auto Prompt Optimizer for further refinement; if passes, move on to the next sample(with higher $c$). The cycle repeats until no error remains or the budget is reached, yielding a self-improved prompt.

**Learning objective.** The parameters $\psi$ are learned by minimizing a hybrid objective that balances the KL penalty and the bounded surrogate loss:

$$\min_{\psi} \ R(\psi) \ + \ \lambda \, \mathbb{E}_{(\tilde{x},\tilde{y})\sim q_\psi}\big[L\big(f(p,\tilde{x}),\, y\big)\big], \tag{1}$$

Note that, we penalize deviations from the true label distribution using the Kullback–Leibler divergence term $R(\psi) = \mathrm{KL}\big(q_\psi(y) \,\|\, p^*(y)\big)$, scaled by a factor $\lambda^{-1}R(\psi)$ during training.

**Progressive difficulty.** To address tasks of varying difficulty, we introduces a progressive difficulty parameter $c$ where $c \in \{1,...,n\}$ so that prompts could be tested on gradually more challenging examples. This allows the prompts to progressively improve and generalize effectively across task of increasing difficulty. Since $q_\psi$ is conditioned on $c$, a single latent template $(z,y)$ can therefore yield $n$ difficulty-aligned variants

$$\{\tilde{x}^{(1)}, \cdots, \tilde{x}^{(n)}\} = \{q_\psi(z,y,1), \cdots, q_\psi(z,y,n)\}$$

For curriculum generation, an ordered sequence $c_1 < \cdots < c_n$ is sampled and feeds the output of the previous level back into the generator,

$$\tilde{x}^{(1)} = q_\psi\big(z,\, y,\, c_1\big), \tilde{x}^{(2)} = q_\psi\big(h_\phi\big(x^{(1)}\big),\, y,\, c_2\big), \ldots, \tilde{x}^{(n)} = q_\psi\big(h_\phi\big(x^{(n-1)}\big),\, y,\, c_n\big),$$

where $h_\phi$ is a summarizer that distills the previous sample into a new latent cue, allowing semantic depth to accumulate across levels. The sequence $c_1 < c_2 < \cdots < c_n$ guarantees monotone growth of problem difficulty, providing a rich gradient of difficulty for the prompt to learn from.

### 3.2 Auto Prompt Optimizer

After each new synthetic instance is calibrated, the Auto Prompt Optimizer probes the current prompt, identifies the weaknesses, and repairs them before the next instance is drawn. This stage builds a prompt that is both robust, suitable, and generalizable for specific tasks.

**Accuracy score.** At iteration $t \in \{1, \ldots, M\}$, the optimizer improves the current prompt $p^{(t)}$ using the feedback collected from synthetic log $D_t = \{(\tilde{x}_j, \tilde{y}_j)\}_{j=1}^t \subseteq \mathcal{X} \times \mathcal{Y}$. For any prompt $p$ and set $\mathcal{A} \subseteq \mathcal{X} \times \mathcal{Y}$, we define

$$s_{\mathcal{A}}(p) \ = \ \frac{1}{|\mathcal{A}|} \sum_{(\tilde{x},\tilde{y})\in\mathcal{A}} \mathbb{I}\big[f(p,\tilde{x}) = \tilde{y}\big], \tag{2}$$

$\mathbb{I}[\cdot]$ is the indicator function that evaluates to 1 if the prompt's output matches the target label, and 0 otherwise.

**Step 1: Error analysis.** We first evaluate $p^{(t)}$ on the whole set and collect the current error slice

$$\mathcal{E}^{(t)} \ = \ \big\{(\tilde{x},\tilde{y})\in D \mid f(p^{(t)},\tilde{x}) \neq \tilde{y}\big\}. \tag{3}$$

If $\mathcal{E}^{(t)} = \varnothing$, the prompt already "covers" all unseen cases, therefore, we terminate and return $p^* = p^{(t)}$; otherwise, we proceed to the next step.

**Step 2: Recommendation.** A reflection module $\mathcal{R}_\varphi$ inspects $\mathcal{E}^{(t)}$, $(\tilde{x}, \tilde{y})$, $p^{(t)}$, and $f(p^{(t)}, \tilde{x})$ and produces a textual-patch suggesting how to modify the prompt: $\Delta^{(t)} = \mathcal{R}_\varphi(p^{(t)}, \mathcal{E}^{(t)}, (\tilde{x}, \tilde{y}), f(p^{(t)}, \tilde{x}))$. This summarizes why the prompt failed and how it can be amended(e.g., clarify/revise instructions, drop distracting details).

**Step 3: Targeted refinement.** A prompt editor $\mathcal{U}_\theta$ applies the patch $\Delta^{(t)}$ to a revised prompt $\tilde{p}^{(t)}$ in order to fix the current error: $\tilde{p}^{(t)} = \mathcal{U}_\theta(\Delta^{(t)}, p^{(t)}, \mathcal{E}^{(t)})$.

**Local confirmation.** We then test revised prompt $\tilde{p}^{(t)}$ only on the current errors: if $s_{\mathcal{E}^{(t)}}(\tilde{p}^{(t)}) < 1$, some errors still remain. In this case, we make the revised prompt as new baseline prompt by setting $p^{(t)} \leftarrow \tilde{p}^{(t)}$, updating $\mathcal{E}^{(t)}$, and repeating Step 2 to generate more sufficient patch $\Delta^{(t)}$; otherwise, proceed to global confirmation.

**Global confirmation.** Solving the local error slice is not enough-we must ensure that revised prompt "covers" all seen cases. Therefore, we evaluate $\tilde{p}^{(t)}$ on the entire synthetic history collected seen so far by $s_{D_t}(\tilde{p}^{(t)})$. During evaluation, if $\mathcal{E}^{(t)} \neq \varnothing$ at any previous data, we treat them as new error set and sent them back to step 2 with new $\mathcal{E}^{(t)}$ to fix the current error. If $\mathcal{E}^{(t)} = \varnothing$, we accept the revision, set $p^{(t+1)} = \tilde{p}^{(t)}$, draw the next synthetic example, and restart from Step 1 until $t = M$.

**Convergence guarantee.** Because $s_D(p^{(t)})$ is non-decreasing and bounded above by 1, the process stops at most $M$ successful corrections or the user-chosen cap $T_{\max}$. The final output $p^* = \arg\max_{0 \le t \le T} s_D(p^{(t)})$ achieves perfect coverage ($s_{D_T}(p^*) = 1$) whenever it is attainable within the budget.

By iteratively applying this feedback-driven process, it systematically refines prompts to improve clarity, adaptability, and overall performance, making the framework highly generalizable across tasks and domains.

## 3.3 THEORETICAL GUARANTEE

Since one of our goals in SIPDO is to demonstrate that, data augmentation, a popular branch of performance improvement in deep learning, can also be used in prompt optimization context, we aim to offer similiar performance guarantees as done in previous data augmentation literature (Wang et al., 2022; Chen et al., 2020; Dao et al., 2019).

**Assumptions.** We first offer the assumptions that we need for the theoretical guarantees.

**A1 (Label-preservation)** *For all $\psi \in \Psi$ and for any $(x, y)$, the generator's conditional satisfies* $\Pr_{q_\psi}[\tilde{y} = y \mid \tilde{x} \overset{g}{\leftarrow} (x, y)] = 1$.
We require the generator never flips the ground-truth label of the base example it is derived from (it may, however, hallucinate novel inputs as long as their labels match the intended classes). In practice, because LLMs sometimes assign unexpected labels, we first generate the label $\tilde{y}$ and then sample $\tilde{x}$ conditioned on that label. For tasks and domains where producing valid synthetic data is difficult, we apply a three-voter check: three expert agents independently verify each generated item for label–input consistency and basic factual correctness.

**A2 (Approximate maximizer).** *Let $\psi^\star = \arg\max_{\psi \in \Psi} \mathbb{E}_{q_\psi} L(f(p, \tilde{x}), \tilde{y}) - \lambda^{-1} R(\psi)$, The inner-loop training of the generator attains a value at most $\varepsilon$ below this supremum.*
A perfect maximizer would be ideal but is infeasible; we only need the learned generator to be good enough—within $\varepsilon$ of optimal. The residual $\varepsilon$ directly appears in the bound.

**A3 (Uniform convergence).** (Wang et al., 2022) *For every prompt $p$, the empirical loss deviates from its population counterpart by at most $q(|\mathcal{P}|, n, \delta)$ with probability $1 - \delta$., where a standard form of $q(|\mathcal{P}|, n, \delta)$ is $\tilde{O}\left(\sqrt{\frac{\log |\mathcal{P}| + \log(1/\delta)}{N}}\right)$.*
PAC(probably approximately correct) guarantee: empirical performance generalizes provided $n$ is large enough.

| MMLU Subject | Method | 1st iteration | 2nd iteration | 3rd iteration | Final Result (Comparative Acc.) |
|---|---|---|---|---|---|
| College Computer Science | TextGrad | 85 | 88 | 86 | 89($\downarrow$ 4.0) |
| | M-TextGrad | 85 | 87 | 87 | 89($\downarrow$ 4.0) |
| | REVOLVE | 85 | 88 | 89 | 90($\downarrow$ 3.0) |
| | **SIPDO** | – | – | – | **93** |
| Machine Learning | TextGrad | 84.8 | 87.5 | 82.1 | 88.4($\downarrow$ 5.4) |
| | M-TextGrad | 85.5 | 85.4 | 85.3 | 85.0($\downarrow$ 8.8) |
| | REVOLVE | 85.7 | 86.6 | 85.7 | 88.4($\downarrow$ 5.4) |
| | ANN | – | – | – | 90.1($\downarrow$ 3.7) |
| | **SIPDO** | – | – | – | **93.8** |
| College Biology | TextGrad | 95.1 | 97.2 | 95.1 | 96.5($\downarrow$ 0.0) |
| | REVOLVE | 96.5 | 96.1 | 97.2 | 96.5($\downarrow$ 0.0) |
| | **SIPDO** | – | – | – | **96.5** |

Table 1: Results on MMLU Machine Learning, College Computer Science, and College Biology subject by GPT-4o, demonstrating SIPDO's effectiveness on different subjects

**A4 (Alignment of risks).** *For any prompt $p$ and generator $\psi$,*

$$\mathbb{E}_{q_\psi} L(f(p, \tilde{x}), \tilde{y}) \leq \mathbb{E}_S L(f(p, x), y) + \lambda^{-1} R(\psi).$$

The KL penalty controls how far the generator may wander: if it manufactures rare-label outliers, $R(\psi)$ increases and the bound tightens. We can verify that $q_\psi(y)$ is always absolutely-continuous w.r.t. $p^*(y)$; KL is then finite and the inequality follows from the classical Donsker–Varadhan variational formula.

**A5 (Surrogate link).** *The 0–1 loss is upper-bounded by the surrogate loss:* $\mathbb{1}\{f(p, x) \neq y\} \leq L(f(p, x), y)$.
This is needed in order to translate guarantees on the differentiable training loss to the classification error(e.g. cross-entropy, hinge, logistic).

**Theorem 3.1 Regularised Worst-case Data Generation.** Under Assumptions A1-A5, for any fixed prompt $p \in \mathcal{P}$, with probability at least $1 - \delta$ over the draw of the training set, we have

$$\underbrace{\sup_{\psi \in \Psi} \mathbb{E}_{q_\psi} \mathbb{1}\{f(p, \tilde{x}) \neq \tilde{y}\}}_{\substack{\text{population} \\ \text{worst-case error}}} \leq \underbrace{\frac{1}{n} \sum_{i=1}^n L\big(f(p, x_i), y_i\big)}_{\text{empirical risk}} + \underbrace{\lambda^{-1} R(\psi^\star)}_{\substack{\text{KL penalty of} \\ \text{hardest generator}}} + \varepsilon + q\big(|\mathcal{P}|, n, \delta\big). \quad (2)$$

**Practical implication.** The inequality states that *if* the empirical loss of the prompt is low, *and* no generator can inflate that loss without paying a high KL tax, *then* even a hypothetically all-powerful adversary (generator) cannot cause the prompt to misclassify more than the RHS. Selecting a larger $\lambda$ tightens the KL tax, thus lowering the worst-case error but potentially harming accuracy—precisely the robustness–performance trade-off observed empirically in Experiments Section 4. In addition, detailed proof of theorem can be found in Appendix F.

# 4 EXPERIMENTS

## 4.1 EXPERIMENTAL SETUP

We use following baseline methods are for comparison across the different datasets and benchmarks: Chain of Thought (CoT) (Suzgun et al., 2022) guides models through explicit step-by-step reasoning; Automatic Prompt Engineer (APE) (Wang et al., 2023) refines prompts via Monte Carlo search and model feedback; PromptAgent (Zhou et al., 2022) also uses Monte Carlo Tree Search to iteratively improve prompts; Neuro-Symbolic (Pan et al., 2023) converts LLM outputs into structured forms for rule-based inference; TextGrad (Yuksekgonul et al., 2024) treats textual feedback as a

first-order gradient for prompt updates; Momentum-Enhanced TextGrad (Yuksekgonul et al., 2024) adds momentum, enlarging updates when feedback aligns; REVOLVE (Zhang et al., 2024) adjusts prompts using the trajectory of model responses as a second-order-style signal; and ANN (Ma et al., 2025) models agent collaboration as a layered neural network.

We test SIPDO on five main datasets across reasoning tasks:

**BIG-Bench.** We include all 4689 instances from six BIG-Bench tasks: Penguins In a Table, Geometric Shapes, Epistemic Reasoning, Object Counting, Temporal Sequences, and Causal Judgment (Srivastava et al., 2022). For these tasks, SIPDO is compared to Chain of Thought (CoT) (Suzgun et al., 2022), Automatic Prompt Engineer (APE) (Wang et al., 2023), and PromptAgent (Zhou et al., 2022).

**Logical Reasoning Tasks.** To assess logical reasoning, we sample 600 examples from the depth-5 subset of ProofWriter with a balanced label distribution (Tafjord et al., 2021), use 204 test examples from FOLIO that require first-order inference over short passages (Han et al., 2024), and select the 500 most challenging 5-hop scenarios from the fictional-character version of PrOntoQA (Saparov & He, 2022). For these tasks, SIPDO is compared to Chain of Thought (CoT) (Suzgun et al., 2022), Neuro-Symbolic (Pan et al., 2023), and REVOLVE (Zhang et al., 2024).

**MMLU (Massive Multitask Language Understanding).** To test expert-level factual knowledge and problem solving in LLMs, we evaluate on MMLU (Hendrycks et al., 2020), focusing on college-level subject areas: Biology (114 instances), Computer Science (100 instances), and Machine Learning (112 instances). For this benchmark, SIPDO is compared with TextGrad (Yuksekgonul et al., 2024), REVOLVE (Zhang et al., 2024), and ANN (Ma et al., 2025).

## 4.2 IMPLEMENTATION

**Data Generation and Prompt Improvements.** We specify a maximum level $c$ and data generated with level of difficulties in prior iterations so that model is aware of the difficulty level of each previous example. To illustrate this, we provide a detailed example of generating a Causal Judgment QA pair from BIG-Bench below.

---

### Causal Judgment Prompt Template

`[System Input]:` You are an expert in generating logical causal-attribute questions and answers. Your task is to generate one pair of causal attributions or causation. One is a causation statement, and another is a non-causation statement.

`[User Input]:`

**Guidelines:**
1. Make sure that the data generated is different.
2. Use clear and direct words in the question, avoiding overly complex phrasing, trickiness, or ambiguity.
3. Do not make the logic in the statement overly complicated; the statement and question should be understood at a fair level.
4. Be creative and diverse. Only follow the logic and flow of the given examples, but be creative and diverse with the content and not limited to the given example.
5. Only one data instance needs to be generated each time.
6. Generate only one output: the generated content should strictly follow the output format from the examples below.
7. Difficulty should increase with each iteration with total difficulty level: {max difficulty level} (current difficulty level: {c}).
8. Make sure the generated data is sufficient and robust without any errors, especially logic.
9. The generated samples are suppose to challenge the model's ability to reason and answer the question correctly.

**Past generated samples:** {Generated data with difficulty}

**Below are the Examples with expected data format:**
{True Data 1}
{True Data 2}

---

| Model | Method | Accuracy (%) | | | | | | Avg. Acc. (%) (Comparative Acc.) |
|---|---|---|---|---|---|---|---|---|
| | | Penguins | Geome. | Epistemic | Obj. Count | Temporal | Causal | |
| GPT-4o | CoT | 79.8 | 79.1 | 79.3 | 85.2 | 98.0 | 67.8 | 81.5(↓ 7.6) |
| | APE | 84.8 | 65.3 | 84.8 | 86.0 | 99.2 | 74.0 | 82.4(↓ 6.7) |
| | PromptAgent | 96.1 | **83.0** | **91.6** | 88.2 | 98.4 | 77.8 | **89.2**(↑ 0.1) |
| | **SIPDO** | **96.4** | 82.2 | 86.3 | **91.1** | **99.3** | **79.0** | 89.1 |
| GPT-4o-mini | CoT | 75.8 | 68.6 | **85.2** | 81.5 | 94.9 | 63.6 | 78.3(↓ 9.0) |
| | APE | 83.7 | 44.5 | 81.6 | 86.3 | 97.2 | 75.6 | 78.2(↓ 9.1) |
| | PromptAgent | 89.8 | 72.0 | 86.0 | 84.3 | 94.6 | 84.6 | 85.2(↓ 2.1) |
| | **SIPDO** | **92.1** | **73.2** | 85.1 | **87.5** | **98.0** | **88.0** | **87.3** |
| Gemini-1.5-flash | CoT | 70.4 | 68.3 | 85.5 | 90.1 | 94.0 | 66.8 | 79.2(↓ 3.7) |
| | APE | 37.6 | 49.4 | **88.8** | 84.7 | **99.4** | 69.4 | 71.6(↓ 11.3) |
| | PromptAgent | 67.4 | **70.3** | 81.6 | 86.3 | 94.2 | 67.9 | 78.0(↓ 4.9) |
| | **SIPDO** | **77.3** | 68.9 | 87.0 | **92.3** | 98.4 | **73.2** | **82.9** |
| Gemini-1.5-pro | CoT | **81.8** | 59.1 | 82.6 | **92.8** | **98.9** | 61.5 | 79.5(↓ 3.9) |
| | APE | 40.2 | 56.6 | 88.7 | 78.6 | 86.0 | 65.7 | 69.3(↓ 14.1) |
| | PromptAgent | 73.6 | 58.3 | 83.8 | 72.6 | 98.4 | 74.2 | 76.8(↓ 6.6) |
| | **SIPDO** | 79.3 | **64.3** | **89.3** | 91.3 | 98.0 | **78.3** | **83.4** |

Table 2: Results on BIG-Bench tasks across multiple LLMs. SIPDO consistently outperforms standard prompting baselines (CoT, APE, PromptAgent) across most tasks and models, demonstrating generalization and effectiveness of the prompt optimization by synthetic data feedback.

We set the difficulty budget at $c = 10$ for all benchmarks except Penguins In a Table and Geometric Shapes from BIG-Bench, where $c = 25$ accommodates their complex reasoning. The number of training iterations is tied to the difficulty level, with $t = c$ to ensure progressively harder samples. The model temperature is set to 0.5 for data generation to maintain coherence. We fix the target label $\tilde{y}$ and prompt the model to generate a matching question for data validity. For challenging MMLU benchmarks, three expert agents review each generated item, and only those with unanimous approval are passed to the auto-prompter. This minimizes hallucinations and ensures accurate question–answer pairs. Examples of synthetic BIG-Bench Causal Judgment data at different difficulty levels are shown in Appendix E.

For further prompt optimization, we set the model temperature to 0.0 during error analysis and improvement steps to ensure deterministic, high-quality outputs. However, we use a higher temperature of 0.7 for generating recommendations, which encourages the model to produce more diverse and creative suggestions. This combination stabilizes the error analysis and prompt improvement phases while simultaneously enhancing the breadth and quality of recommendations. The full prompt-improvement process for BIG-Bench Penguins In a Table is in Appendix D.

**Geometric Data Generation.** Constructing complex or irregular shapes exceeds the limits of few-shot methods, so we introduce three safeguards for SVG path generation in the geometry task: (1) *precision normalization*—each coordinate is rounded to two decimal places, preventing floating-point drift that makes downstream parsers miscount line (`L`) and arc (`A`) commands; (2) *template-guided retrieval*—a retriever selects an SVG path template whose instruction pattern matches the target shape (e.g., "4 L" for a rectangle, "1 A" for a sector), and the generator perturbs only the vertex coordinates, ensuring syntactic correctness while adding variety; (3) *reverse-generation check*—because the shape label is known in advance, a rule-based decoder parses the generated path, tallies `L/A` commands, and rejects any sample whose inferred label disagrees. Examples appear in Appendix E.

## 4.3 RESULTS AND ANALYSIS

We tested SIPDO across various LLMs with temperature of 0.0 on all benchmarks, including BIG-Bench, FOLIO, PrOntoQA, ProofWriter, and three subjects from MMLU. Specifically, we ran SIPDO on GPT-4o, GPT-4o-mini, Gemini-1.5-flash, and Gemini-1.5-pro, so that all synthetic-data calls and prompt refinements in result were driven by the same model.

**MMLU.** We first evaluate SIPDO on three subjects from the MMLU benchmark: Machine Learning, College Biology, and College Computer Science. As shown in Table 1, SIPDO achieves the highest results in all three subjects, while TextGrad and REVOLVE tie in the Biology subject. Specifically,

| Tasks | GPT-4o | | | | | GPT-4o-mini | | | | |
|---|---|---|---|---|---|---|---|---|---|---|
| | Vanilla | Neuro-S | CoT | REVOLVE | **SIPDO** | Baseline | Neuro-S | CoT | REVOLVE | **SIPDO** |
| ProofWriter | 58.5 | **81.6** | 72.3 | 54.0 | 79.6 | 52.6 | **79.7** | 61.8 | 48.6 | 79.3 |
| FOLIO | 71.2 | 79.2 | 72.6 | 65.7 | **83.2**(↑ 4.0) | 51.2 | 73.2 | 69.3 | 62.8 | **81.1**(↑ 7.9) |
| PrOntoQA | 80.4 | 85.2 | 95.6 | 85.4 | **96.3**(↑ 0.7) | 74.6 | 79.3 | 89.3 | 83.4 | **91.3**(↑ 2.0) |
| Average | 70.0 | 82.0 | 80.2 | 68.4 | **86.4**(↑ 4.2) | 59.5 | 77.4 | 73.5 | 64.9 | **83.9**(↑ 6.5) |

Table 3: Performance(%) on ProofWriter, FOLIO, and PrOntoQA by Neuro-Symbolic, CoT, RE-VOLVE, SIPDO, and Baseline Prompting methods across GPT-4o and GPT-4o-mini.

| Model | PENGUINS | GEOME. | EPISTEMIC | OBJ.CNT. | TEMPORAL | CAUSAL | Avg. |
|---|---|---|---|---|---|---|---|
| GPT-4o | 73.2 (↓ 24.1%) | 68.1 (↓ 17.2%) | 81.9 (↓ 5.1%) | 53.8 (↓ 40.9%) | 97.0 (↓ 2.3%) | 68.4 (↓ 13.4%) | 73.7 (↓ 17.3%) |
| GPT-4o-mini | 69.6 (↓ 24.4%) | 47.5 (↓ 35.1%) | 80.0 (↓ 6.0%) | 39.9 (↓ 54.4%) | 92.1 (↓ 6.0%) | 67.4 (↓ 23.4%) | 66.1 (↓ 24.3%) |

Table 4: Accuracy (%) after removing the difficulty gradient. Numbers in parentheses show the absolute drop (↓) relative to the performance with difficulty gradient placed.

we set up the SIPDO pipeline using the GPT-4o model, except for the GPT-4o-mini during the testing phase, in order to provide greater insight into the reasoning weaknesses of LLMs.

**BIG-Bench.** We then evaluate SIPDO on six BIG-Bench tasks As shown in Table 2, GPT-4o, GPT-4o-mini, and Gemini-1.5-flash demonstrate particularly strong performance in Temporal Reasoning, Object Counting, Penguins In a Table, and Causal Judgment. While Geometric Shapes exhibits comparable accuracy across GPT-4o and GPT-4o-mini, SIPDO achieves the highest overall accuracy across all LLMs except for GPT-4o, trailing PromptAgent by only 0.1%, yet still demonstrating that LLMs benefit from synthetic data generation for reasoning improvements, whereas other methods primarily rely on existing datasets. To further illustrate the generated different difficulty data in Causal Judgment tasks as iteration and difficulty increase, the actual logical turns display a monotonic trend in figure 2.

**FOLIO, PrOntoQA, and ProofWriter.** We further test SIPDO on FOLIO, PrOntoQA, and ProofWriter, assessing the methods' ability to perform structured logical reasoning. As shown in Table 3, SIPDO outperforms all approaches on FOLIO and PrOntoQA and achieves the highest average accuracy. In PrOntoQA, SIPDO surpasses all methods, demonstrating its capability to generate structured logical proofs. Similarly, for FO-LIO, SIPDO outperforms Neuro-Symbolic, CoT, and RE-VOLVE, further validating its effectiveness in formal logic inference.

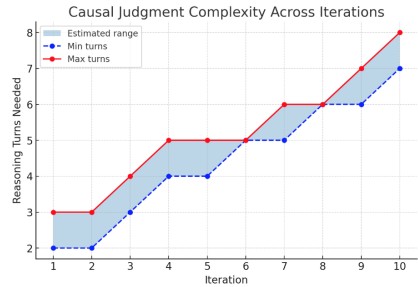

Figure 2: Generated BIG-Bench Causal Judgment task in different difficulties

While neuro-symbolic reasoning remains the best performer on ProofWriter, SIPDO achieves highly competitive results on ProofWriter, trailing by only 0.4% on GPT-4o-mini and 2% on GPT-4o, underscoring its strong adaptability to structured reasoning tasks. Crucially, unlike neuro-symbolic approaches that rely on predefined rule-based datasets, SIPDO is trained entirely on generated synthetic data, demonstrating the effectiveness of LLM-driven data augmentation for enhancing logical inference across diverse reasoning benchmarks. SIPDO outputs a revised prompt that surpasses baselines, validating its effectiveness for prompt design and performance gains. All generated prompts appear in Appendix B.

## 4.4 ABLATION STUDY

**Difficulty Gradient.** To assess the contribution of the difficulty gradient, we conduct an ablation study by comparing without difficulty level. As Table 4 shows, every BIG-Bench sub-task suffers

when the difficulty gradient is absent. On average, GPT-4o loses $17.3\%$ accuracy, while the weaker GPT-4o-mini drops $24.3\%$, confirming that smaller models depends even more on the difficulty gradient. The steepest declines appear on tasks Object Counting (40.9 % and 54.4 %) and Geometric Shapes (17.2 % and 35.1 %). Even comparatively simple tasks—Temporal Sequences and Epistemic Reasoning—still lose up to 6 %. We also provide a detailed analysis of how different difficulty levels affect performance in Table 5. These results indicate that within the OpenAI model family, the weaker model is more sensitive to the absence of a difficulty gradient and the benefit of a progressive difficulty schedule becomes more pronounced. Without this gradient, generated prompts tend to be shorter and easier, often failing to capture complex reasoning patterns (details in Appendix C).

**One-Shot Extremes.** We experimented with replacing the step-wise difficulty gradient by a one-shot extremes sampler that tells the generator to create the most unusual examples. On our synthetic suites this shortcut delivered no measurable gain. The "extreme" samples were either solved instantly or only slight perturbations of original cases, leaving the optimizer with no fresh errors to exploit. We suspect the idea will pay off in real-world corpora (e.g. financial statements) where genuine edge cases abound and can expose blind spots that our synthetic tasks do not capture.

## 5 CONCLUSION

We presented SIPDO, a data-centric framework that converts augmentation into a live feedback signal for prompt optimization. A generator creates progressively harder examples, and an auto-prompt optimizer uses them to expose and correct prompt weaknesses. This coupling produces consistent accuracy gains across diverse reasoning benchmarks, outperforming several leading baselines. Beyond these empirical results, SIPDO shows how data-generation strategies and prompt optimization can reinforce one another, linking ideas from curriculum learning and adaptive optimization with LLM practice. Further investigation on domain specific corpora such as financial filings and clinical notes and exploration of fully automated variants that refine prompts through continuous model feedback will clarify SIPDO's broader value and extend its principles to new settings.

## ACKNOWLEDGMENT

This work was partially supported by the National Artificial Intelligence Research Resource (NAIRR) Pilot under awards NAIRR250400 and NAIRR240283, and Standing Up to POTS.

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

# A    DETAILED ABLATION TABLE STUDIES

In this section, we provide more comprehensive and detailed ablation studies that demonstrate our studies and findings.

| Task | Difficulty | GPT-4o-mini | | | | GPT-4o | | | |
|---|---|---|---|---|---|---|---|---|---|
| | | Run Time (s) | Cost ($) | Tokens | Acc. (%) | Run Time (s) | Cost ($) | Tokens | Acc. (%) |
| Penguins | 5 | 217.43 | 0.0130 | 67269 | 83.7 | 58.55 | 0.1089 | 35426 | 91.3 |
| | 10 | 1078.47 | 0.0929 | 527358 | 92.59 | 561.05 | 1.2100 | 410807 | 93.33 |
| | 15 | 2226.68 | 0.2174 | 1250646 | 92.59 | 2392.23 | 6.4528 | 2275582 | 96.0 |
| | 20 | 19753.99 | 2.6698 | 16695611 | 93.33 | 3452.01 | 7.8194 | 2759306 | 96.67 |
| Epistemic | 5 | 49.07 | 0.0027 | 14297 | 75.33 | 119.28 | 0.1401 | 42985 | 78.0 |
| | 10 | 686.18 | 0.0420 | 227607 | 76.0 | 555.86 | 0.6814 | 224480 | 79.0 |
| | 15 | 1471.69 | 0.1149 | 640713 | 80.33 | 1440.52 | 2.0357 | 695678 | 83.67 |
| | 20 | 3279.38 | 0.2583 | 1483605 | 80.0 | 2885.80 | 4.0457 | 1418053 | 84.0 |
| Geometric | 5 | 51.70 | 0.00369 | 19079 | 35.33 | 106.06 | 0.0955 | 27548 | 50.0 |
| | 10 | 690.22 | 0.03237 | 144212 | 49.17 | 560.95 | 0.4726 | 133780 | 67.67 |
| | 15 | 1326.86 | 0.06710 | 296540 | 72.22 | 964.28 | 0.9158 | 268912 | 80.0 |
| | 20 | 2503.59 | 0.20610 | 571178 | 78.06 | 1334.71 | 1.4882 | 450575 | 87.78 |
| causal_judgement | 5 | 103.80 | 0.0064 | 35013 | 58.42 | 149.56 | 0.1611 | 52383 | 67.95 |
| | 10 | 899.33 | 0.0699 | 385346 | 66.66 | 156.22 | 0.2761 | 101715 | 81.6 |
| | 15 | 2066.64 | 0.2017 | 1152961 | 79.67 | 534.45 | 0.6921 | 237519 | 82.42 |
| | 20 | 443.71 | 0.0388 | 236561 | 67.89 | 3033.81 | 4.8447 | 1738332 | 79.49 |
| temporal | 5 | 35.40 | 0.0016 | 7650 | 94.0 | 39.27 | 0.0283 | 7813 | 99.67 |
| | 10 | 147.00 | 0.0066 | 26538 | 97.5 | 647.34 | 0.8005 | 267894 | 99.33 |
| | 15 | 2166.98 | 0.1876 | 1039249 | 99.67 | 1432.08 | 2.1206 | 736935 | 99.33 |
| | 20 | 1518.23 | 0.1187 | 630972 | 99.64 | 2235.50 | 3.5084 | 1255135 | 99.66 |
| object counting | 5 | 190.52 | 0.0122 | 60765 | 91.0 | 115.30 | 0.1660 | 49776 | 97.0 |
| | 10 | 1094.08 | 0.0684 | 363795 | 92.0 | 512.40 | 0.7296 | 232588 | 98.0 |
| | 15 | 2592.44 | 0.2069 | 1148406 | 96.0 | 1396.33 | 1.8831 | 624918 | 99.33 |
| | 20 | 5078.75 | 0.3968 | 2260081 | 96.0 | 2848.81 | 4.3167 | 1459739 | 99.33 |

Table 5: SIPDO's cost (time, money, tokens) and accuracy across tasks and difficulty levels for GPT-4o-mini and GPT-4o.

| Model | Method | Penguins | | Epistemic | | Geometric | | Temporal | | Causal Judgment | | Object Counting | |
|---|---|---|---|---|---|---|---|---|---|---|---|---|---|
| | | Run Time (s) | Acc. (%) | Run Time (s) | Acc. (%) | Run Time (s) | Acc. (%) | Run Time (s) | Acc. (%) | Run Time (s) | Acc. (%) | Run Time (s) | Acc. (%) |
| GPT-4o | PromptAgent | 18300 | 96.1 | 23675 | 91.6 | 30639 | 83.0 | 40399 | 98.4 | 18232 | 77.8 | 37453 | 88.2 |
| | SIPDO | 2392 | 96.4 | 556 | 86.3 | 1335 | 82.2 | 647 | 99.3 | 156 | 79.0 | 1094 | 91.1 |
| GPT-4o-mini | PromptAgent | 9398 | 89.8 | 24150 | 86.0 | 46012 | 72.0 | 48561 | 94.6 | 15097 | 84.6 | 41977 | 84.3 |
| | SIPDO | 2227 | 92.1 | 686 | 85.1 | 2504 | 73.2 | 147 | 98.0 | 899 | 88.0 | 512 | 87.5 |

Table 6: Run time and accuracy across tasks for SIPDO and PromptAgent

| Task | Cost in time (s) | | Cost in money ($) | | Performance (%) | |
|---|---|---|---|---|---|---|
| | Remove voters | With voters | Remove voters | With voters | Remove voters | With voters |
| College Biology | 763 | 912 | 0.0167 | 0.0221 | 95.14 | 96.5 |
| Machine Learning | 375 | 680 | 0.0096 | 0.7974 | 76.79 | 93.8 |
| College Computer Science | 432 | 1311 | 0.0104 | 1.8080 | 88.00 | 93.0 |

Table 7: Impact of SIPDO's voters on cost and performance across tasks.

| Task | Accuracy (%) | | | Cost in time (s) | | |
|---|---|---|---|---|---|---|
| | w/o generator | w/o error analysis | w/o revisor | w/o generator | w/o error analysis | w/o revisor |
| College Biology | 95.83 | 95.83 | 95.14 | 331 | 677 | 314 |
| Machine Learning | 90.18 | 84.82 | 77.68 | 767 | 1622 | 353 |
| College Computer Science | 94.00 | 90.00 | 89.00 | 2405 | 2852 | 2861 |

Table 8: Ablation results on accuracy and time cost when removing each component of SIPDO.

## B    OPTIMZED PROMPTS FOR DIFFERENT TASKS

In this section, we demonstrate optimized prompts by Chain-of-Thought (CoT), Automatic Prompt Engineering (APE), PromptAgent, and SIPDO with Accuracys respectively.

Table 9: Comparison of Optimized Prompts for **Object Counting task**, including CoT, APE, PromptAgent, and SIPDO

| Approach | Optimized Prompt | Accuracy |
|---|---|---|
| CoT | Your task is to count the total number of objects mentioned in the question. Follow these simple steps to ensure accurate counting: **Steps to Follow:** 1. **Identify Items**: Read the question carefully and list all objects mentioned. 2. **Count Quantities**: For each item, check if a quantity is provided. If no quantity is mentioned, assume it is one. 3. **Add Totals**: Add up the quantities of all items to calculate the total count. 4. **Verify the Total**: Double-check to ensure no item is missed or counted twice. **Example:** - Question: "Count the apples, oranges, and bananas. There are 2 apples, 1 orange, and 3 bananas." - Step 1: Identify items: apples, oranges, bananas. - Step 2: Count quantities: 2 apples, 1 orange, 3 bananas. - Step 3: Add totals: 2 + 1 + 3 = 6. - Step 4: Verify: All items are accounted for, total is 6. - **Output**: "The total count is 6." Use this step-by-step method for every question to ensure accurate and clear results. | 0.928 |
| APE | Calculate the overall total of all items even those spoken in groups. | 0.863 |
| PromptAgent | Carefully examine the provided information. Identify and catalog each mentioned item, ensuring that explicitly stated quantities are accurately recorded. If no quantity is specified for an item, assume it as a single unit. However, for items with defined quantities, count each unit separately and include it in the total. If collective terms or categories are mentioned, break them down into their individual components and associate each with its stated count. When computing the total for such categories, ensure that the sum reflects all individual units rather than just the number of groups or types. Each item should be counted independently, but related items belonging to a common category should be grouped together, with their specific quantities contributing precisely to the final total. Avoid assumptions regarding the classification or nature of items—adhere to standard, widely accepted definitions. Finally, summarize the count by explicitly listing the quantity of each identified item or category, and provide a comprehensive total of individual units rather than just category counts, unless otherwise specified. | 0.882 |
| SPIDO | Task Requirements: The task involves counting the total number of objects listed in a question. Each distinct object should be considered as part of the total count, regardless of its type or variation. The output should be formatted correctly as specified. Problem Rule Application: Identify all items listed in the question. Count each item exactly once, regardless of type, to determine the total number of objects. Ensure accuracy by verifying that all listed items have been included in the final count. Provide the final result in the required format: The number should be presented in both word form and numerical form, separated by a comma (e.g., "nine, 9"). No extra symbols, characters, or explanations should be included. Judgment Criteria: (Strictly follow these rules) Complete Identification: Extract and recognize all objects in the given list. Do not overlook any item mentioned in the question. Accurate Counting: Each item must be counted exactly once. Ensure no items are omitted or double-counted. Verification Process: Double-check the list to confirm that all objects are included. Cross-verify the final count to avoid errors. | 0.923 |

Table 10: Comparison of Optimized Prompts for **Penguins In A Table task**, including CoT, APE, PromptAgent, and SIPDO

| Approach | Optimized Prompt | Accuracy |
|---|---|---|
| CoT | You are tasked with answering questions about a table of penguins and their attributes. Use step-by-step reasoning to ensure accuracy in calculations and comparisons. The table is as follows: "' Name, Age, Height (cm), Weight (kg) Louis, 7, 50, 11 Bernard, 5, 80, 13 Vincent, 9, 60, 11 Gwen, 8, 70, 15 "' **Reasoning Steps for Each Question:** 1. Identify the target attribute (age, height, or weight) and the type of operation (comparison, ranking, filtering). 2. Extract the relevant rows or columns based on the question's requirements. 3. Perform the required operation step-by-step using the extracted data. 4. Clearly summarize the answer based on the operation's result. Example Workflow: - Question: "Who is the tallest penguin?" - Step 1: Identify the target attribute: Height. - Step 2: Extract the height values and corresponding names: [(Louis, 50), (Bernard, 80), (Vincent, 60), (Gwen, 70)]. - Step 3: Find the maximum height: Bernard (80 cm). - Step 4: Output the result: "Bernard is the tallest penguin with a height of 80 cm." Follow this workflow for every question to ensure clarity and correctness. | 0.818 |
| APE | Carefully scrutinize the provided table or tables. Understand the query in relation to the information given. Pinpoint the pertinent data and carry out the vital computations or comparisons to determine the right answer from the given choices. | 0.848 |
| PromptAgent | Answer questions about a table of penguins and their attributes, considering both the penguin table and any additional relevant tables. Please provide step-by-step reasoning for your answers, and ensure to clarify any criteria used for filtering or sorting data. Here is a table where the first line is a header and each subsequent line is a penguin: name, age, height (cm), weight (kg) Louis, 7, 50, 11 Bernard, 5, 80, 13 Vincent, 9, 60, 11 Gwen, 8, 70, 15 For example: the age of Louis is 7, the weight of Gwen is 15 kg, the height of Bernard is 80 cm. What is the name of the last penguin sorted by alphabetic order? Options: (A) Louis (B) Bernard (C) Vincent (D) Gwen (E) James **Instructions**: 1. List the names of the penguins. 2. Sort the names alphabetically and present the sorted list clearly. 3. Identify the last name in the sorted list and indicate the corresponding option letter from the provided options. 4. If the last name does not match any of the options, select the name that is closest to the last name in the original list of penguins. At the end, show the answer option bracketed between ¡answer¿ and ¡/answer¿. | 0.961 |
| SIPDO | Answer questions about a dynamic, comprehensive table of penguins and their attributes that allows penguins and other animals to be added and removed. Perform calculations and comparisons based on the questions asked. Read the question carefully to determine which attribute is being compared (age, height, weight). When comparing an attribute, extract the name and that attribute, and then compare, ignoring the other attributes. Ensure the extracted value is from the correct column corresponding to the requested attribute. When using the table, align the data so that the first number is age, the second is height, and the third is weight. Understand the question correctly, find the key words from it, and then perform calculations or comparisons based on the key words The current table is as follows: Name, Age, Height (cm), Weight (kg) Louis, 7, 50, 11 Bernard, 5, 80, 13 Vincent, 9, 60, 11 Gwen, 8, 70, 15 **Question Rules to Apply:** - Identify the rows or columns that meet the specified conditions. - Retrieve the value of the required attribute from the identified rows or columns. When we modify this table (by adding new penguins or removing existing penguins or adding giraffes), we first confirm whether the information we added is a penguin or a giraffe, and then solve the problem of comparing, ranking, and filtering based on attributes between penguins or giraffes, depending on the problem. | 0.964 |

Table 11: Comparison of Optimized Prompts for **Geometric Shapes task**, including CoT, APE, PromptAgent, and SIPDO

| Approach | Optimized Prompt | Accuracy |
|---|---|---|
| CoT | Your task is to identify the geometric shape represented by the given SVG path data. Follow these steps to ensure accuracy: 
 **Steps to Identify the Shape:** 1. **Check for 'A' Instructions**: – If the path contains 'A', determine: • **Circle**: 2 or more 'A' instructions. • **Sector**: 1 'A' instruction. 2. **Count 'L' Instructions**: – If there are no 'A' instructions, count the 'L' instructions to determine the polygon's shape: • **Line**: 2 'L'. • **Triangle**: 3 'L'. • **Rectangle**: 4 'L'. • **Pentagon**: 5 'L'. • **Hexagon**: 6 'L'. • **Heptagon**: 7 'L'. • **Octagon**: 8 'L'. • **Kite**: 4 'L'. 3. **Provide the Shape Name**: Output only the name of the shape (e.g., "circle", "triangle", "hexagon"). **Example:** – Input: '"M 10 10 L 20 10 L 20 20 L 10 20 Z"' – Step 1: No 'A' instructions. – Step 2: Count 'L' instructions: 4 'L'. – Step 3: Shape is a **Rectangle**. – **Output**: "rectangle". 
 Use this step-by-step process for all inputs to determine the correct shape. | 0.791 |
| APE | Determine the shape each SVG path element is drawing, then pair it with the corresponding letter from the available choices. In this case, C symbolizes hexagon, G is for pentagon, I signifies sector, and B stands for heptagon. | 0.650 |

*Continued on next page*

Table 11: Comparison of Optimized Prompts for **Geometric Shapes task** (continued)

| Approach | Optimized Prompt | Accuracy |
|---|---|---|
| PromptAgent | Analyze the SVG path data to identify the geometric shape it represents. Follow these comprehensive and refined steps to ensure accurate identification: 
 1. **Holistic Path Closure**: Determine if the path forms a closed shape by checking if the last point connects back to the starting point. If multiple 'M' commands are present, analyze the segments collectively to identify any closed loops. Treat the entire path as a single entity for thorough analysis. 2. **Segment and Side Analysis**: Identify the types of segments used in the path: – **Line Segments**: Count the number of distinct line segments to determine the number of sides. Ensure accurate counting by considering all segments collectively. – **Arc Segments**: For paths using the 'A' command, note that these represent elliptical arcs. Pay attention to the parameters to distinguish between circles and ellipses. 3. **In-depth Geometric Properties**: – For line segments, analyze the relative lengths of sides and angles between them. Consider properties such as parallel sides, equal side lengths, and right angles to distinguish between different types of polygons. Evaluate the overall shape formed by all segments. – For arc segments, examine the parameters of the 'A' command: • Check if the radii are equal, which indicates a circle. • If the radii differ, consider the shape as an ellipse. 4. **Shape Identification and Classification**: Use the geometric properties to classify the shape: – For polygons, identify specific types like rectangles, kites, and trapezoids based on their properties. Pay special attention to the number of sides and the relationships between them. Consider the entire path as a single shape to ensure accurate classification. – For arcs, determine if the shape is a circle or an ellipse based on the radii. 5. **Options Selection and Interpretation**: Choose the most appropriate shape from the given options. Consider multiple interpretations of the path data, especially when multiple 'M' commands are present, to ensure accurate classification. If the path represents multiple shapes, prioritize the most complex or relevant shape. 6. **Ambiguity Resolution**: In cases where the path data could represent multiple shapes, provide a rationale for selecting the most complex or relevant shape. Consider the context and any additional information that might influence the classification. 7. **Visual Verification**: If possible, visualize the path to confirm the identified shape. This step can help resolve any remaining ambiguities and ensure the accuracy of the classification. 8. **Iterative Refinement**: If the initial classification is uncertain, revisit the analysis steps to refine the identification. Consider alternative interpretations and re-evaluate the geometric properties. 9. **Contextual Considerations**: Take into account any contextual information or additional data that might influence the shape classification, especially in ambiguous cases. 
 Provide your answer by selecting the correct option and enclosing it within '¡answer¿' and '¡/answer¿' tags. 
 **Example:** – SVG Path: 'path d="M 8.10,55.86 L 1.74,25.57 L 12.08,23.40 L 18.44,53.69 L 8.10,55.86"' Analysis: The path forms a closed quadrilateral with opposite sides parallel and equal, indicating a rectangle. Answer: '¡answer¿H¡/answer¿' – SVG Path: 'path d="M 16.33,5.98 A 8.87,8.87 275.02 1,0 14.78,23.64 A 8.87,8.87 275.02 1,0 16.33,5.98"/' Analysis: The path uses elliptical arcs with equal radii, forming a closed loop, indicating a circle. Answer: '¡answer¿A¡/answer¿' | 0.830 |

*Continued on next page*

Table 11: Comparison of Optimized Prompts for **Geometric Shapes task** (continued)

| Approach | Optimized Prompt | Accuracy |
|---|---|---|
| SIPDO | Given the following SVG path data: "input" and options, identify the geometric shape it represents and provide **ONLY** the name of the shape as the 'target'. 
**Task Requirements:** 1. Count the instructions in the SVG path 2. Judge the shape of the graphic according to the judgment criteria 3. Provide the exact name of the shape as output. 
You need to count how many instructions **L** are in the SVG path: 
**Problem Rule Application:** 1. Visualize the path data to understand the overall structure. 2. Find out whether there is instruction **A** in the instruction. If so, determine whether it is a circle or a sector according to the number of instructions **A**. If not, determine how many sides it is 3. For polygons, pay attention to the number of edges to identify the shape. The following are the number of instructions corresponding to different shapes: – **triangle**: 3 L – **rectangle**: 4 L – **hexagon**: 6 L – **pentagon**: 5 L – **octagon**: 8 L – **heptagon**: 7 L – **kite**: 4 L – **line**: 2 L – **circle**: 2 or more A – **sector**: 1 A 
**Judgment criteria:** (Please strictly abide by this rule) – No need to pay attention to "M" instructions – !! First identify whether there is an instruction "A" in the SVG path. If so, first determine whether it is a circle or a sector. – !! If there is no instruction "A", determine the number of sides of the polygon based on the instruction "L". A polygon with *n* sides requires *n* "L" instructions. (Please strictly abide by this rule) | 0.822 |

Table 12: Comparison of Optimized Prompts for **Causal Judgment tasks**, including CoT, APE, PromptAgent, and SIPDO

| Approach | Optimized Prompt | Accuracy |
|---|---|---|
| CoT | Task: Respond to inquiries about causal attribution by identifying the key causes and their contributions to the outcome. Follow the steps below to ensure accurate and clear reasoning: 
**Steps to Analyze Causation:** 1. **Identify Key Entities**: Read the question carefully and highlight the specific entities or factors being discussed. 2. **Determine Relevant Causes**: Analyze the context to identify immediate and incidental causes contributing to the outcome. - Immediate causes: Directly lead to the outcome. - Incidental causes: Indirectly influence the outcome but may still contribute. 3. **Evaluate Interactions**: Consider how multiple causes might interact to produce the observed effect (e.g., synergy or independent contributions). 4. **Provide the Answer**: Clearly state the primary and secondary causes, as well as their roles in creating the outcome. Avoid unsupported assumptions. 
Use this structured reasoning approach to analyze each inquiry and provide a clear and logical explanation. | 0.678 |
| APE | For each situation, decide if the result was caused deliberately or not. If the individual or party behind the event was aware of the potential result and chose to go ahead, select 'Yes'. If they didn't intend the result to happen, even if they knew it could possibly occur, select 'No'. | 0.756 |

*(Continued on next page)*

*(Continued from previous page)*

| Approach | Optimized Prompt | Accuracy |
|---|---|---|
| PromptAgent | When addressing questions about causal attribution, ensure a comprehensive analysis by considering both individual and collective actions that contribute to an outcome. Clearly differentiate between necessary and sufficient causes, and recognize that multiple causes can simultaneously contribute to an outcome. Emphasize the importance of understanding both general and specific intentions, especially when outcomes are unintended. Define "intentional" actions as those where the actor or group had control over maintaining or altering the conditions necessary for the outcome, even if the specific result was not desired. Address scenarios where unintended consequences arise from intentional actions, and provide answers that reflect a nuanced understanding of how different elements interact to produce a result. Use diverse examples to illustrate key concepts like "direct causation," "simultaneity," and "unintended consequences," ensuring a balanced consideration of necessary and sufficient causes. Simplify complex scenarios by breaking them down into clear, manageable components, and provide definitions or examples of key terms to guide your analysis. Additionally, clarify definitions of key terms such as "necessary," "sufficient," "intentional," and "unintended consequences" to ensure precise understanding. Highlight the importance of interactions between multiple causes, especially in complex scenarios, and offer strategies for analyzing scenarios where simultaneity is crucial. Explore the nuances of intentional actions and unintended consequences more deeply, and encourage the use of diverse examples to illustrate different aspects of causation. Pay special attention to the role of individual actions in maintaining necessary conditions and the distinction between collective and individual causation. Emphasize that in collective decision-making, the outcome can be intentional if it aligns with the group's goals, even if individual members disagree. Reinforce the distinction between necessary and sufficient causes, ensuring the model understands that necessary causes alone do not determine the outcome. Clarify that following a protocol does not remove intentionality if the outcome aligns with organizational priorities. Highlight that intentionality can be attributed if the outcome was a foreseeable consequence of the action, regardless of individual opposition. | 0.846 |
| SIPDO | Task Requirements Determine whether a given event (cause) directly leads to another event (effect). Assess the causal relationship based on logical reasoning, ensuring a clear and definitive answer. The final output must be only "Yes" or "No", strictly adhering to the required format. Problem Rule Application Identify the cause and effect within the question. Assess necessity: Determine if the cause is essential for the effect to occur. Evaluate causation: If the cause did not happen, would the effect still occur? If the effect only happens when the cause is present, then the cause directly leads to the effect. If the effect can still happen independently, then the relationship is not causal. Judgment Criteria Direct Causation: If the cause directly leads to the effect and is a necessary condition, answer "Yes". If the effect would not have occurred without the cause, answer "Yes". Example: "Dropping a glass caused it to shatter." → Yes. No Direct Causation: If the effect can occur without the cause, answer "No". If the cause is only correlated but not necessary, answer "No". Example: "Wearing a red shirt caused the stock market to rise." → No. Verification Process: Check whether the absence of the cause results in the absence of the effect. Ensure logical consistency in the causal assessment. | 0.880 |

Table 13: Comparison of Optimized Prompts for **Epistemic task**, including CoT, APE, PromptAgent, and SIPDO

| Approach | Optimized Prompt | Accuracy |
|---|---|---|
| CoT | Task: Analyze the logical relationship between a given premise and hypothesis. Your goal is to determine if the premise guarantees the truth of the hypothesis. Choose one of the following answers: 'entailment' or 'non-entailment'.

**Steps to Follow:** 1. **Understand the Premise and Hypothesis**: Carefully read the premise and hypothesis to identify the key information in both statements. 2. **Analyze the Logical Relationship**: Determine whether the information in the premise confirms the truth of the hypothesis. - If the premise logically supports and guarantees the hypothesis, choose 'entailment'. - If the premise does not confirm the hypothesis, or if there is uncertainty, choose 'non-entailment'. 3. **Provide the Answer**: Based on your analysis, output the correct answer ('entailment' or 'non-entailment').

Use this step-by-step approach for all premise and hypothesis pairs to ensure accurate reasoning. | 0.855 |
| APE | Determine whether the hypothesis is directly implied by the premise or not. If the premise's statement is a direct claim or conviction of the individual mentioned in the hypothesis, choose 'entailment'. However, if the premise is formed on the belief or supposition of someone other than the subject in the hypothesis, opt for 'non-entailment'. | 0.888 |
| PromptAgent | Determine the relationship between two sentences by evaluating whether the first sentence provides direct or logically implied evidence for the second. Choose from the options 'entailment' or 'non-entailment'.

Consider the following: - **Entailment**: The first sentence directly or through logical implication confirms the truth of the second sentence, even if it involves a chain of beliefs or perceptions, as long as the chain logically supports the hypothesis. - **Non-entailment**: The first sentence does not confirm the truth of the second sentence, often involving unsupported assumptions, beliefs, or suspicions that do not logically lead to the hypothesis.

Guidelines for Analysis: 1. **Clarify Belief Chains and Logical Implications**: Understand how belief chains work and when they logically support the hypothesis. Pay attention to verbs indicating beliefs, assumptions, or suspicions (e.g., "thinks," "assumes," "suspects") versus those indicating direct evidence (e.g., "learns," "knows," "remembers"). Consider how these verbs interact in belief chains and what they imply about the subject's own beliefs. 2. **Evaluate Direct and Implied Evidence**: Determine if the premise provides direct or logically implied evidence for the hypothesis, considering how belief chains can logically support the hypothesis. Recognize that indirect beliefs about another person's recognition can imply one's own belief about a situation, especially when the belief chain is logical and straightforward. 3. **Consider Perspective and Source of Information**: Note any differences in perspective or source of information (e.g., who remembers or assumes something) and how these perspectives contribute to the logical implication of the hypothesis. 4. **Conduct a Comprehensive Analysis**: Use a step-by-step approach to ensure all relevant details and logical implications are considered in the analysis. Balance the emphasis on direct evidence with the recognition of logical implications from indirect beliefs.

Example: Premise: "Charlotte thinks that Richard recognizes that a boy is standing in a pool getting splashed with water." Hypothesis: "Charlotte thinks that a boy is standing in a pool getting splashed with water." Options: (A) entailment (B) non-entailment

Analysis: 1. **Understanding the Premise**: The premise indicates that Charlotte thinks Richard recognizes a specific situation involving a boy in a pool. 2. **Understanding the Hypothesis**: The hypothesis states that Charlotte thinks a boy is in a pool getting splashed with water. 3. **Assessing the Relationship**: The premise implies that Charlotte has a belief about the situation (through Richard's recognition), which logically supports the hypothesis. Charlotte's belief about Richard's recognition suggests she also believes in the situation's occurrence. 4. **Conclusion**: The relationship is one of entailment because Charlotte's belief about Richard's recognition logically implies her belief in the situation.

Therefore, the correct answer is:
`<answer>A</answer>`
Identify the relation between the following premises and hypotheses, choosing from the options 'entailment' or 'non-entailment'. At the end, show the answer option bracketed between `<answer>` and `</answer>`. | 0.916 |

*(Continued from previous page)*

| Approach | Optimized Prompt | Accuracy |
|---|---|---|
| SIPDO | Task Requirements:
Analyze a given premise (primary sentence) and determine whether it fully supports the truth of a hypothesis (subsequent sentence). Classify the relationship as either "Entailment" or "Non-Entailment" based on the logical and factual connections between the two. Provide the classification only as the final output. Problem Rule Application:
Entailment:
The premise explicitly confirms the hypothesis with clear, direct evidence. No additional information, assumptions, or interpretations are required to validate the hypothesis. Non-Entailment:
The premise does not fully or explicitly confirm the hypothesis. If there is ambiguity, uncertainty, or missing logical links, label it as Non-Entailment. Judgment Criteria: (Strictly follow these rules)
Language of Uncertainty:
Words like "assumes," "believes," "thinks," "feels," "suspects" indicate subjectivity and should not be considered definitive proof. These terms suggest a possibility rather than an explicit factual connection. Specific vs. General Statements:
A specific premise (e.g., mentioning a "full face mask") does not necessarily contradict a general hypothesis (e.g., referencing a "mask" in general). However, if the premise is too general to confirm the specific claim, classify as Non-Entailment. Objective Reasoning:
Only use the logical and factual ties within the given statements. Do not rely on external knowledge, assumptions, or interpretations unless directly supported by the premise. Decision Process:
Determine whether the premise fully supports the hypothesis without needing extra inference → Entailment. If the premise only partially supports or fails to confirm the hypothesis → Non-Entailment. | 0.893 |

Table 14: Comparison of Optimized Prompts for **Temporal task** including CoT, APE, PromptAgent, and SIPDO.

| Approach | Optimized Prompt | Accuracy |
|---|---|---|
| CoT | Your task is to determine the available time slot for an event, based on a schedule of occupied times. Follow these steps to ensure accuracy:
**Steps to Identify Free Time Slots:** 1. **List Occupied Periods**: Organize all occupied time slots in chronological order. 2. **Find Gaps**: Identify gaps between the occupied periods where no activities are scheduled. 3. **Check Constraints**: Ensure that the free time slots fall within operational constraints (e.g., facility closing times). 4. **Select the Slot**: Choose the correct free time slot that satisfies all criteria. **Output Result Format:** - Present the selected free time slot in a clear format, such as "Xpm to Ypm" or "Xam to Yam".
Use this step-by-step method to ensure that the identified time slot is accurate and does not overlap with any occupied periods. | 0.989 |
| APE | Identify the period when the individual was unnoticed and had the possibility to visit the specified place before its closing time. | 0.994 |

*(Continued on next page)*

*(Continued from previous page)*

| Approach | Optimized Prompt | Accuracy |
|---|---|---|
| PromptAgent | Analyze the timeline of events to determine possible time frames during which certain events could have occurred, even if they were not explicitly observed. Start by constructing a comprehensive timeline, clearly listing all observed and unobserved time slots. Identify gaps where the subject is unobserved, ensuring these gaps fit within any given constraints, such as opening and closing times. Emphasize the importance of constraints by verifying them after identifying potential gaps. Use a step-by-step reasoning approach to systematically evaluate all available information, and include a final review to check for potential errors or overlooked details before finalizing the answer. Define key terms like "unobserved" and "constraints" to ensure clarity in the task requirements. Provide examples to illustrate the reasoning process and expected output format, guiding the model in analyzing timelines and identifying possible time frames for unobserved events. Additionally, incorporate a checklist to ensure all steps are followed, and highlight common pitfalls to avoid during the analysis. Finally, include a summary of the reasoning process to reinforce understanding and ensure the model's conclusions are well-supported.

To further enhance the model's performance, include additional examples that cover a wider range of scenarios and constraints, such as overlapping time slots or multiple constraints. Provide explicit guidance on handling complex constraints and ambiguous information. Incorporate interactive feedback mechanisms to help the model learn from mistakes and improve over time. Ensure the prompt is concise and focused, avoiding unnecessary repetition while maintaining clarity and comprehensiveness. Additionally, introduce a section for handling exceptions or unusual cases, offering strategies for dealing with incomplete or conflicting data. This will help the model adapt to a broader range of real-world scenarios and improve its robustness in timeline analysis tasks. | 0.984 |
| SIPDO | **Task Requirements:** Determine the possible time period during which an event could have occurred, based on a detailed schedule of occupied times. Your goal is to identify the correct time slot that fits all the provided criteria without any overlap.
**Problem Rule Explanation:** 1. Analyze the schedule to identify all time slots during which the person is occupied. 2. Determine the available time slots by identifying gaps between these occupied periods. 3. Consider any additional constraints, such as closing times, that may limit the available time slots.
**Problem Rule Application:** - List all the occupied time slots chronologically. - Identify gaps between these occupied slots where the person is free. - Ensure that the free time slots do not conflict with constraints like closing times.
**Result Verification:** - Confirm that the identified time slot is completely free and adheres to any constraints. - Double-check against all occupied periods to ensure there is no overlap. - Avoid selecting time slots that are partially occupied or overlap with any scheduled activities.
**Output Result Format:** - Present the correct time slot in a straightforward manner, using the format "Xpm to Ypm" or "Xam to Yam" as appropriate. - Ensure the output is clear and free of any extraneous symbols or text.
**Common Mistakes to Avoid:** - Do not include time slots that extend beyond the closing time of the facility. - Avoid selecting time slots that overlap with any scheduled activities. - Ensure the selected time slot is entirely free and does not partially overlap with any occupied period.
**General Rules and Analysis:** - Identify all occupied periods and list them chronologically. - Look for gaps between these periods where the person is not scheduled for any activity. - Verify that these gaps fall within any operational constraints, such as closing times. - Ensure the selected time slot is entirely free and does not overlap with any occupied periods.
By following these guidelines, you can accurately determine the available time slot for the event in question. Avoid errors by ensuring that the selected time slot is entirely free and does not overlap with any occupied periods. | 0.993 |

## C  OPTIMIZED PROMPTS WITHOUT DIFFICULTY SCALING IN SYNTHETIC DATA

Table 15: Optimized Prompts Without Difficulty Scaling in Synthetic Data

| Tasks | Optimized Prompt | Accuracy |
|---|---|---|
| Penguins | You are provided with two tables containing data about penguins and giraffes. Your task is to focus solely on the giraffe data to answer a specific question regarding the tallest giraffe.
**Penguin Data:**
— Name — Age — Height (cm) — Weight (kg) — ——————————————————
————-— — Louis — 7 — 50 — 11 — — Bernard — 5 — 80 — 13 — — Vincent — 9 — 60 — 11 — — Gwen — 8 — 70 — 15 — — James — 12 — 90 — 12 —
**Giraffe Data:**
— Name — Age — Height (cm) — Weight (kg) — ——————————————————
————-— — Jody — 5 — 430 — 620 — — Gladys — 10 — 420 — 590 — — Marian — 2 — 310 — 410 — — Donna — 9 — 440 — 650 —
**Task Requirements:** 1. Identify the tallest giraffe based on the height provided in the Giraffe Data table. 2. Provide the weight of the tallest giraffe in kilograms.
**Problem Rule Explanation:** - Review the height values for each giraffe listed in the Giraffe Data table. - Compare these height values to determine which giraffe is the tallest.
**Problem Rule Application:** - Examine the height values for the giraffes: - Jody: 430 cm - Gladys: 420 cm - Marian: 310 cm - Donna: 440 cm - Identify that Donna is the tallest giraffe at 440 cm. - Retrieve the corresponding weight of Donna, which is 650 kg.
**Result Verification:** - Ensure that you have considered all entries in the Giraffe Data table. - Confirm that the weight you provide corresponds to the giraffe identified as the tallest.
**Output Result Format:** - Provide your answer in the following format: - "Weight of the tallest giraffe: [Weight in kg]"
—
**Example Output:** - "Weight of the tallest giraffe: 650"
— | 0.732 |
| Geometry | " Given the following input: ""input"", you must provide ONLY the correct value for the 'target'.
**Rules:** 1. Do NOT provide any explanations. 2. Do NOT provide any sentences, text, or words other than the 'target' value. 3. The answer must be the exact value contained in the ""target"" and any unauthorized additions are prohibited." | 0.681 |
| Object Counting | "**Task Requirements:** - Determine the total number of fruits by accurately identifying and counting each type listed in the question.
**Problem Rule Explanation:** - The task involves listing and counting each fruit mentioned. - Each fruit should be counted as one unless a specific quantity is provided.
**Problem Rule Application:** - Carefully read through the list to identify all items that are fruits. - Count each fruit once unless otherwise specified with a different quantity. - Avoid including any non-fruit items or miscounting due to misinterpretation of the list.
**Result Verification:** - Re-examine the list to ensure all fruits have been correctly identified and counted. - Verify that the total count reflects only the fruits listed, with no errors in inclusion or exclusion.
**Output Result Format:** - Provide the total number of fruits in both word and numeral forms, such as: [""ten"", ""10""]. - Ensure the output is clear and free from special symbols or formatting errors." | 0.538 |

*(Continued from previous page)*

| Tasks | Optimized Prompt | Accuracy |
|---|---|---|
| Causal Judgment | Analyze the scenario to determine if the described action directly caused the outcome. Provide a definitive 'Yes' or 'No' answer based on a logical assessment of the causal relationship as described in the scenario. **Problem Rule Explanation:** A causal relationship exists when an action directly leads to an outcome without other factors influencing the result. The outcome should not occur without the action. Avoid assumptions and base your analysis solely on the information provided. **Problem Rule Application:** - Identify the key action and the resulting outcome within the scenario. - Determine if the outcome is a direct result of the action, ensuring no additional factors are at play. - Evaluate whether the outcome would still occur without the initial action, focusing on the explicit roles, responsibilities, and conditions mentioned. - Avoid external assumptions and concentrate on the details provided in the scenario. **Result Verification:** - Confirm that the action directly causes the outcome, with no interference from other factors. - Ensure the outcome logically follows from the action, considering the context and rules provided. - Review the scenario for any overlooked details that could affect the causal link, ensuring a comprehensive analysis. **Output Result Format:** - Answer 'Yes' if the action directly causes the outcome, with the outcome being a direct consequence of the action. - Answer 'No' if there is no direct causal relationship or if other factors could have contributed to the outcome. | 0.684 |
| Temporal | **Task Requirements:** Determine the available time slots for an unscheduled activity within a given daily schedule, ensuring these slots do not conflict with scheduled events and comply with any facility operating hours. **Problem Rule Explanation:** 1. Review the entire schedule to identify all events and their specific time frames. 2. Identify gaps between these events or after the last scheduled event to find potential time slots for the unscheduled activity. 3. Consider any additional constraints, such as facility operating hours, to ensure the proposed time slot is feasible. **Problem Rule Application:** - List all scheduled events with their respective time frames. - Identify gaps between these events or available time after the last scheduled event. - Ensure that the identified time slots comply with any additional constraints, like facility operating hours. **Result Verification:** - Confirm that the identified time slots do not overlap with any scheduled events. - Verify that the time slots fall within the facility's operating hours. **Output Result Format:** Present the time range in a clear and concise format, such as "Xpm to Ypm" or "Xam to Yam", ensuring clarity and precision. **Example Application:** Given the schedule: - Breakfast: 8am to 9am - Business meeting: 9am to 11am - Art gallery: 11am to 1pm - Lunch: 1pm to 2pm - Cinema: 3pm to 5pm - Dinner party: 6pm to 8pm - Gym closes at 10pm Determine the available time for the gym: - Identify the gaps: 5pm to 6pm and 8pm to 10pm. - Ensure these time slots do not overlap with scheduled events and are within the gym's operating hours. - The correct answer is "5pm to 6pm" and "8pm to 10pm", as they fit within the gym's operating hours and do not overlap with any scheduled events. | |

## D    SELF-IMPROVEMENT PROMPT TEMPLATES

We provide an example of a self-improvement prompt template in the BIG-Bench Penguins In a Table task: Error analysis, Improvement recommendations, and prompt refinement.

---

**Error Analysis Prompt Template**

You are an expert at analysing why LLM answer is wrong in Penguins in a table reasoning task.

Your task is to give a concise and general description of the reasoning mistake from the current prompt that caused the mistake with the provided question, generated answer, ground truth, and current prompt below:
Question :{synthetic question}
Model generated answer: {LLM generated answer}
Ground truth: {true answer}
Current prompt: {current prompt}

1. **Misinterpretation of the Question**: The model may have misinterpreted the question, focusing on the structure of the data rather than the specific request for the height of the tallest penguin. This could lead to confusion and an irrelevant answer.

2. **Inattention to Numerical Data**: The model might have overlooked the numerical values provided in the table, failing to recognize that it needed to compare the heights of the penguins to determine the tallest one.

3. **Irrelevant Output Generation**: The answer "rectangle" does not relate to the context of the question. This suggests that the model may have generated a response based on unrelated patterns or associations rather than the specific data presented.

4. **Lack of Contextual Understanding**: The model may not have fully grasped the context of the data table, leading to a failure in recognizing that the question was asking for a specific value derived from the table.

5. **Failure to Process Tabular Data**: The model might struggle with processing tabular data effectively, which can lead to incorrect conclusions or irrelevant outputs when asked to analyze such formats.

---

**Improvement Recommendation Prompt Template**

You are an expert in giving recommendations for optimizing current prompts. The goal is to output reasonable and decent suggestions on how to revise the current prompt to solve the encountered issue.

Question :{synthetic question}
Model generated answer: {LLM generated answer}
Ground truth: {true answer}
Current prompt: {current prompt}
Generated error analysis: {error analysis from previous step}

Some recommendation examples in Penguins In A Table task(But don't be limited to these):
- Clarify the question to emphasize the need for numerical comparison.
- Provide explicit instructions to focus on extracting specific values from the data.
- Ensure the model is trained to recognize and process tabular data more effectively.
- Avoid ambiguity in the question to prevent misinterpretation of the request.

**Prompt Refinement Template**

You are an expert in revising prompt engineering and refinement in tables related to reasoning tasks.

Goal: Create a revised prompt that fixes the failure without overfitting – keep it generic.
Question :{synthetic question}
Model generated answer: {LLM generated answer}
Ground truth: {true answer}
Current prompt: {current prompt}
Refinement recommendation: {recommendation from previous step}

Tasks:
- Give a better prompt, which can avoid all the problems that have occurred.
- Summarize the logic based on the question and correct answer, and summarize the mistakes that should be avoided based on the question and generated wrong answer.
- Do not modify the prompt based on the specific problem, but modify the prompt based on the cause of the error. The modified prompt should be able to give some regular hints and logical revisions.
- You should specify the output format according to the correct answer. The output should not contain any special symbols.
- You can summarize the characteristics of each option and reverse the answer based on the characteristics.
- It can analyze the general rules of the problem, which helps the model understand how the problem is solved.
- Prompts should be planned according to the following categories: task requirements, problem rule explanation, problem rule application, result verification, and output result format.

# E EXAMPLES OF GENERATED DATA

Table 16: Examples of Generated Data for BIG-Bench tasks

| Tasks | Generated Data |
|---|---|
| Causal Judgment | Question: In a small town, there is a bakery that makes the best pastries. Every morning, the bakery opens at 7:00 am, and a line of customers forms outside. The bakery owner has a rule that only one person can enter at a time to maintain order. One day, two customers, Alice and Bob, arrive at the same time. Alice follows the rule and waits outside, but Bob ignores the rule and enters the bakery while another customer is still inside. The bakery becomes overcrowded, and a shelf of pastries falls over, ruining the day's batch. Did Bob cause the pastries to be ruined?
Answer: Yes |
| Geometry | Question:`This SVG path element <path d="M 50.00,30.00 L 66.18,35.09 L 72.45,50.00 L 66.18,64.91 L 50.00,70.00 L 33.82,64.91 L 27.55,50.00 L 33.82,35.09 L 50.00,30.00"/>`
Answer: Polygon |
| Object Counting | Question: I have two violins, a drum, a piano, a flute, and a trumpet. Additionally, I have a cat, a rabbit, a dog, a chicken, and a goat. How many musical instruments do I have?
Answer: 4 |
| Epistemic | Question: Premise: Olivia suspects that Ethan recognizes that a group of musicians gather in a park, tuning their instruments as the sun sets behind the city skyline. Hypothesis: Ethan recognizes that a group of musicians gather in a park, tuning their instruments as the sun sets behind the city skyline.
Answer: non-entailment |
| Temporal | Question:Today, Alex attended several events. Between what times could he have gone to the gym? We know that: Alex had breakfast at 8am. He attended a meeting from 9am to 11am. He was seen at the art gallery from 11am to 1pm. He had lunch with friends from 1pm to 2pm. He was at the cinema from 2pm to 4pm. He visited his grandmother from 4pm to 6pm. The gym closes at 10pm. Between what times could Alex have gone to the gym? 12pm to 1pm, 9am to 10am, 6pm to 10pm, 4pm to 5pm
Answer: 6pm to 10pm |
| Penguins | Question: Here is a table where the first line is a header and each subsequent line is a penguin: name, age, height (cm), weight (kg)
Louis, 7, 50, 11
Bernard, 5, 80, 13
Vincent, 9, 60, 11
Gwen, 8, 70, 15. For example: the age of Louis is 7, the weight of Gwen is 15 kg, the height of Bernard is 80 cm. Which is the youngest penguin??
Answer: Bernard |

Table 17: Examples of Generated Causal Judgement Data for BIG-Bench tasks

| Difficulty Level/Iteration | Generated Data |
| --- | --- |
| 1 | In a small town, there is a bakery known for its delicious pastries. The bakery owner, Mr. Thompson, has a strict policy that only his trained staff can bake the pastries. However, one day, a customer named Sarah, who has some baking experience, decides to sneak into the kitchen while the staff is busy. She bakes a batch of pastries without permission. The next morning, Mr. Thompson discovers that the pastries are not up to his usual standards. Did Sarah cause the pastries to be of poor quality? |
| 2 | In a bustling restaurant, the head chef, Chef Maria, has a rule that only she and her sous chefs are allowed to create new dishes. One evening, a waiter named Tom, who has a passion for cooking, decides to experiment and prepares a new dish during a slow shift. The next day, Chef Maria tastes the dish and finds it lacks flavor and presentation. Did Tom cause the dish to be of poor quality |
| 3 | In a high-tech company, the lead engineer, Ms. Johnson, has a strict protocol that only certified technicians are allowed to perform software updates on the main server. One afternoon, a junior intern named Alex, eager to impress, decides to bypass the protocol and updates the software without authorization. The next day, the server experiences significant downtime due to compatibility issues. Did Alex cause the server downtime? |
| 4 | In a prestigious university, the dean, Dr. Roberts, has a clear policy that only faculty members are allowed to supervise student research projects. However, a graduate student named Emily, frustrated with the slow progress of her project, decides to take matters into her own hands and conducts an experiment without faculty approval. When the results are presented, they are deemed unreliable and flawed. Did Emily cause the research results to be unreliable? |
| 5 | In a renowned art gallery, the curator, Ms. Patel, has a strict guideline that only professional artists can submit their work for exhibition. One day, a passionate art student named Jake, feeling confident in his skills, sneaks in his painting and hangs it on the wall without permission. When the exhibition opens, visitors notice that Jake's painting lacks the finesse and depth of the other artworks. Did Jake cause the negative reception of his painting? |
| 6 | In a large corporate office, the HR manager, Ms. Lee, has a strict policy that only trained personnel are allowed to conduct interviews for new hires. One day, a receptionist named Mark, eager to help, decides to conduct an interview for a candidate without permission. The next day, during the hiring meeting, it becomes clear that the candidate is not a good fit for the company culture. Did Mark cause the poor hiring decision? |
| 7 | In a cutting-edge research lab, the lead scientist, Dr. Nguyen, has a strict rule that only authorized researchers can handle the sensitive equipment. One afternoon, a visiting scholar named Rachel, excited to contribute, decides to use the equipment without permission. The next day, Dr. Nguyen discovers that the data collected is corrupted and unusable. Did Rachel cause the data corruption? |
| 8 | In a competitive publishing house, the editor-in-chief, Ms. Carter, has a strict policy that only experienced editors can approve manuscripts for publication. One day, a new intern named David, eager to make a mark, decides to approve a manuscript without consulting anyone. When the book is published, it is filled with errors and inconsistencies. Did David cause the poor quality of the published book? |
| 9 | In a prestigious law firm, the managing partner, Mr. Stevens, has a strict rule that only licensed attorneys can represent clients in court. One day, a paralegal named Lisa, wanting to prove her capabilities, decides to represent a client during a hearing without authorization. The next day, the judge dismisses the case due to Lisa's lack of legal knowledge and experience. Did Lisa cause the dismissal of the case? |
| 10 | In a leading aerospace company, the project director, Mr. Carter, has a strict rule that only certified engineers can work on the aircraft design. However, his enthusiastic neighbor, Jake, who has no formal training, often offers unsolicited advice on the design process. One day, Jake manages to convince Mr. Carter to incorporate a design feature he believes will improve aerodynamics. Unfortunately, the feature is flawed and leads to a critical failure during a test flight, resulting in the loss of the prototype. Did Jake cause the failure of the prototype? |

## F  DETAILED PROOF OF THEOREM 3.3

**1. Surrogate domination.**  Because the surrogate loss upper-bounds the 0–1 loss point-wise,

$$\sup_{\psi \in \Psi} \mathbb{E}_{q_\psi}\big[\mathbb{1}\{f(p,\tilde{x}) \neq \tilde{y}\}\big] \;\leq\; \sup_{\psi \in \Psi} \mathbb{E}_{q_\psi}\big[L\big(f(p,\tilde{x}),\tilde{y}\big)\big] \;=\; \sup_{\psi \in \Psi} J(\psi),$$

where we abbreviated $J(\psi) := \mathbb{E}_{q_\psi} L\big(f(p,\tilde{x}),\tilde{y}\big)$.

**2. Reduce to the near-optimal generator.**  Let $\psi^\star$ be any generator that $\varepsilon$-maximises the regularised objective,

$$\psi^\star \;=\; \arg\max_{\psi \in \Psi}\Big\{ J(\psi) - \lambda^{-1} R(\psi)\Big\} \quad \text{s.t.} \quad J(\psi^\star) - \lambda^{-1} R(\psi^\star) \;\geq\; \sup_{\psi \in \Psi}\big(J(\psi) - \lambda^{-1} R(\psi)\big) - \varepsilon.$$

Rearranging, $J(\psi) \leq J(\psi^\star) + \lambda^{-1}\big(R(\psi) - R(\psi^\star)\big) + \varepsilon$ for every $\psi$, hence

$$\sup_{\psi \in \Psi} J(\psi) \;\leq\; J(\psi^\star) + \varepsilon.$$

**3. Bound the hard generator via KL.**  Applying the risk-alignment inequality to $\psi^\star$,

$$J(\psi^\star) \;\leq\; \mathbb{E}_{(x,y) \sim P} L\big(f(p,x),y\big) \;+\; \lambda^{-1} R(\psi^\star).$$

**4. Sample–population substitution.**  With probability at least $1 - \delta$ over the draw of the training set,

$$\mathbb{E}_{(x,y) \sim P} L\big(f(p,x),y\big) \;\leq\; \frac{1}{n}\sum_{i=1}^{n} L\big(f(p,x_i),y_i\big) \;+\; q\big(|\mathcal{P}|,n,\delta\big).$$

**Combine.**  Chaining 1-4 we obtain

$$\sup_{\psi \in \Psi} \mathbb{E}_{q_\psi} \mathbb{1}\{f(p,\tilde{x}) \neq \tilde{y}\} \;\leq\; \frac{1}{n}\sum_{i=1}^{n} L\big(f(p,x_i),y_i\big) \;+\; \lambda^{-1} R(\psi^\star) \;+\; \varepsilon \;+\; q\big(|\mathcal{P}|,n,\delta\big),$$

which is exactly the bound claimed in Theorem 3.3. $\qquad\qquad\qquad\qquad\qquad\qquad\qquad\qquad\qquad\qquad\quad$ $\square$

