# OpenReview forum: "SIPDO: Closed-Loop Prompt Optimization via Synthetic Data Feedback"
_ICLR.cc/2026/Conference — ICLR 2026 Poster_

### Official Review · Reviewer_E8bU · 2025-10-31

**Soundness:** 2
**Presentation:** 2
**Contribution:** 2
**Rating:** 4
**Confidence:** 4

**Summary:**

The authors deal with prompt optimization problem for the LLMs, and propose SIPDO, a closed-loop framework that integrates synthetic data generation into prompt optimization. SIPDO consists of two key components: a Data Generator that produces progressively harder synthetic samples to expose prompt weaknesses, and an Auto Prompt Optimizer that refines prompts via error analysis, recommendation, and targeted revision. The framework operates without external supervision, leveraging a feedback loop to iteratively enhance prompt robustness. Experiments on reasoning benchmarks (BIG-Bench, FOLIO, PrOntoQA) and MMLU show that SIPDO outperforms SOTA baselines (APE, PromptAgent, etc.) with multiple LLMs (GPT-4o, Gemini-1.5, etc.).

**Strengths:**

**Originality**
This work demonstrates originality in three folds:  (1) this work integrates synthetic data generation into prompt optimization; (2) the authors introduce to construct synthetic examples with progressive difficulty design to dynamically guide the prompt refinement; (3) The authors provide a theoretical guarantee for the proposed framework to assure the prompt error bounds.

**Quality & Clarity**
The framework is well-structured with self-contained contents. Extensive experiments on a wide range of benchmarks (reasoning, expert knowledge, etc. ) and over 4 LLMs illustrate the effectiveness of the method. Qualitative examples and detailed prompts in appendices provide strong support for this work.

**Significance**
This work addresses a critical point for LLM application --  unreliable prompt performance in dynamic scenarios. The introduced method has the potential to enable robust adaptation of prompts to diverse domains without human annotations. Also, this work advocates to optimize prompt by utilizing synthetic data rather than rule-based datasets in a much more practical manner.

**Weaknesses:**

1. The authors claim no external supervision is used in this method, but the true data including question and answer fed into the "Data Generator" module actually provide the supervision signals for the loop.

2. No quantitative demonstration on the fluctuation of LLMs' performance when the input distribution changes as mentioned in L48?

3. How to define the difficulty level L mentioned in Line 172? No investigation on the influence of this number to the overall performance.

4. The "recommendation" step introduced in Section 3.2 might incorporate the answer of the incorrect cases into the revising recommendation, which can be regarded as a kind of information leakage?

5. Is there any context length control mechanism in the auto prompt optimizer? Since multiple patch $\mathcal{\Delta}^{(t)}$ could be applied to perform prompt revision, and the context could be extended and distracting details could also be dropped dynamically. Longer prompts increase inference latency and may exceed LLM context windows. No analysis on the context length change during the  prompt optimization.

6. Duplicated equation indexes in Line 161 and Line 188. Also no equation indexing in Line 194.

**Questions:**

Several questions and doubts are raised in "Weaknesses" part. I would be willing to change my recommendation according to the authors' response.

---

> ### Author Response · Authors · 2025-11-27
> **Rebuttal By Authors**
>
> We thank the reviewer for the thoughtful and encouraging feedback. We are happy that the reviewer found our feedback loop optimization structure supported by a theoretical guarantee and the quantitative and qualitative analysis from the benchmark and the Appendix to be thorough. We believe the remaining questions can be fully addressed, and we provide a detailed response below.
>
> **1. True data including question and answer fed into the "Data Generator" module actually provide the supervision signals for the loop.**
>
> We thank the reviewer for pointing this out. We apologize for the confusion caused in the paper. We do require minimal predefined task types during synthetic data generation, but what truly ‘no supervision’ means is that the prompt itself can learn from the generated cases while exposing the weakness of the prompt, even the domain shifts. SIPDO is designed to take a minimal seed and autonomously amplify it to generate a large-scale data curriculum for its own refinement loop. In this way, the supervision signal is largely self-generated and self-refined. To reflect this, we have revised in the Line 65-Line 66 to “This feedback loop enables the prompt to evolve continuously over time, adapting to new, previously unseen inputs with minimal supervision for each new scenario individually by providing a general idea of the task type and what needs to be in mind without manually constructing a new prompt.” Also in the abstract, we revised the second-last sentence to “This feedback-driven loop enables systematic improvement of prompt performance with minimal supervision to new tasks.
>
> **2. No quantitative demonstration on the fluctuation of LLMs' performance when the input distribution changes as mentioned in L48.**
>
> We appreciate the reviewer for this observation. We conducted experiments to show the variance and standard deviation across a wide and diverse set of tasks (from BIG-bench and MMLU) as a strong and practical proxy for stability by comparing with 6 methods. The first table shows the variance and standard deviation calculated on all tasks across all LLMs for each method, and SIPDO yields the lowest variance as well as the standard deviation. The second table also indicates SIPDO has the lowest variance and standard deviation in all three MMLU tasks on GPT-4o. Therefore, SIPDO shows the lowest variance and standard deviation in these benchmarks compared with all six methods, which indicates the small fluctuation when the input distribution varies.
>
> | BIG-Bench Task | Variance | Std.  |
> |----------------|----------|-------|
> | **CoT**            | 158.64   | 12.59 |
> | **APE**            | 346.86   | 18.09 |
> | **PromptAgent**    | 105.17   | 9.97  |
> | **SIPDO**          | 102.27   | 9.95  |
>
> | MMLU Task   | Variance | Std. |
> |------------|----------|------|
> | **TextGrad**   | 20.37    | 4.51 |
> | **M-TextGrad** | 8.00     | 2.83 |
> | **Revolve**    | 18.40    | 4.29 |
> | **SIPDO**      | 3.36     | 1.83 |
>
> **3. How to define the difficulty level L mentioned in Line 172? No investigation on the influence of this number to the overall performance.**
>
> We thank the reviewer for this valuable question and suggestion. We apologize for the confusion we caused. The difficulty level L should be n instead, so the difficulty gradient $c_L$ should be $c_n$. $n$ represents the number of iterations, which is the same as the maximum difficulty levels, and $c_1$, $c_2$, …, $c_n$ represent each difficulty level and the number of iterations that SIPDO loops over.
>
> We also conducted experiments to show the trade-off between time cost and performance gains made by changing the difficulty level. We compared different difficulty levels in terms of cost and accuracy across 6 tasks on BIG-Bench. We randomly sampled 300 tasks for Object Counting, Epistemic, Temporal, and Causal Judgment, and used the original number of tasks for Geometry and Penguins In a Table. From the experiments, we observe gains in accuracy as the difficulty level increases, but the improvements appear smaller as the difficulty level approaches 20.

---

> > ### Author Response · Authors · 2025-11-27
> >
> > | Task(By GPT-4o-mini)            | Difficulty | Cost Time(s) | Cost Money($) | Cost Tokens | Accuracy(%) |
> > |----------------|------------|--------------|----------------|-------------|-------------|
> > | Penguins       | 5          | 217.43       | 0.0130         | 67269       | 83.7        |
> > | Penguins       | 10         | 1078.47      | 0.0929         | 527358      | 92.59       |
> > | Penguins       | 15         | 2226.68      | 0.2174         | 1250646     | 92.59       |
> > | Penguins       | 20         | 19753.99     | 2.6698         | 1669561     | 93.33       |
> > | Epistemic      | 5          | 49.07        | 0.0027         | 14297       | 75.33       |
> > | Epistemic      | 10         | 686.18       | 0.0420         | 227607      | 76.00       |
> > | Epistemic      | 15         | 1471.69      | 0.1149         | 640713      | 80.33       |
> > | Epistemic      | 20         | 3279.38      | 0.2583         | 1483605     | 80.00       |
> > | Geometric      | 5          | 51.70        | 0.00369        | 19079       | 35.33       |
> > | Geometric      | 10         | 690.22       | 0.03237        | 144212      | 49.17       |
> > | Geometric      | 15         | 1326.86      | 0.06710        | 296540      | 72.22       |
> > | Geometric      | 20         | 2503.59      | 0.20610        | 571178      | 78.06       |
> > | Causal Judgment| 5          | 103.80       | 0.0064         | 35013       | 58.42       |
> > | Causal Judgment| 10         | 899.33       | 0.0699         | 385346      | 66.66       |
> > | Causal Judgment| 15         | 2066.64      | 0.2017         | 1152961     | 79.67       |
> > | Causal Judgment| 20         | 443.71       | 0.0388         | 236561      | 67.89       |
> > | Temporal       | 5          | 35.40        | 0.0016         | 7650        | 94.00       |
> > | Temporal       | 10         | 147.00       | 0.0066         | 26538       | 97.50       |
> > | Temporal       | 15         | 2166.98      | 0.1876         | 1039249     | 99.67       |
> > | Temporal       | 20         | 1518.23      | 0.1187         | 630972      | 99.64       |
> > | Object Counting| 5          | 190.52       | 0.0122         | 60765       | 91.00       |
> > | Object Counting| 10         | 1094.08      | 0.0684         | 363795      | 92.00       |
> > | Object Counting| 15         | 2592.44      | 0.2069         | 1148406     | 96.00       |
> > | Object Counting| 20         | 5078.75      | 0.3968         | 2260081     | 96.00       |
> >
> > | Task(By GPT-4o)            | Difficulty | Cost Time(s) | Cost Money($) | Cost Tokens | Accuracy(%) |
> > |----------------|------------|--------------|----------------|-------------|-------------|
> > | Penguins       | 5          | 58.55        | 0.1089         | 35426       | 91.3        |
> > | Penguins       | 10         | 561.05       | 1.21           | 410807      | 93.33       |
> > | Penguins       | 15         | 2392.23      | 6.4528         | 2275582     | 96.00       |
> > | Penguins       | 20         | 3452.01      | 7.8194         | 2759306     | 96.67       |
> > | Epistemic      | 5          | 119.28       | 0.1401         | 42985       | 78.00       |
> > | Epistemic      | 10         | 555.86       | 0.6814         | 224480      | 79.00       |
> > | Epistemic      | 15         | 1440.52      | 2.0357         | 695678      | 83.67       |
> > | Epistemic      | 20         | 2885.80      | 4.0457         | 1418053     | 84.00       |
> > | Geometric      | 5          | 106.06       | 0.0955         | 27548       | 50.00       |
> > | Geometric      | 10         | 560.95       | 0.4726         | 133780      | 67.67       |
> > | Geometric      | 15         | 964.28       | 0.9158         | 268912      | 80.00       |
> > | Geometric      | 20         | 1334.71      | 1.4882         | 450575      | 87.78       |
> > | Causal Judgment| 5          | 149.56       | 0.1611         | 52383       | 67.95       |
> > | Causal Judgment| 10         | 156.22       | 0.2761         | 101715      | 81.60       |
> > | Causal Judgment| 15         | 534.45       | 0.6921         | 237519      | 82.42       |
> > | Causal Judgment| 20         | 3033.81      | 4.8447         | 1738332     | 79.49       |
> > | Temporal       | 5          | 39.27        | 0.0283         | 7813        | 99.67       |
> > | Temporal       | 10         | 647.34       | 0.8005         | 267894      | 99.33       |
> > | Temporal       | 15         | 1432.08      | 2.1206         | 736935      | 99.33       |
> > | Temporal       | 20         | 2235.50      | 3.5084         | 1255135     | 99.66       |
> > | Object Counting| 5          | 115.30       | 0.1660         | 49776       | 97.00       |
> > | Object Counting| 10         | 512.40       | 0.7296         | 232588      | 98.00       |
> > | Object Counting| 15         | 1396.33      | 1.8831         | 624918      | 99.33       |
> > | Object Counting| 20         | 2848.81      | 4.3167         | 1459739     | 99.33       |

---

> ### Author Response · Authors · 2025-11-27
>
> **4. The "recommendation" step introduced in Section 3.2 might incorporate the answer of the incorrect cases into the revising recommendation, which can be regarded as a kind of information leakage?**
>
> We appreciate the reviewer for this insightful question. There should not be any information leakage problem. We apologize for the confusion in the recommendation step from the paper. The input to the recommendation step actually contains the original question, LLM-generated answer, ground-truth answer, current prompt, and error analysis, and the LLM is instructed to give recommendations based on all the information provided. We have revised the current recommendation stage to clearly state that the inputs include the error set $\mathcal{E}^{(t)}$, the original question–answer pair $(\tilde{x}, \tilde{y})$, the current prompt $p^{(t)}$, and the LLM-generated answer $f(p^{(t)}, \tilde{x})$. We also provide a recommendation prompt template in Appendix C (page 24).
>
> **5. Is there any context length control mechanism in the auto prompt optimizer? Since multiple patch could be applied to perform prompt revision, and the context could be extended and distracting details could also be dropped dynamically. Longer prompts increase inference latency and may exceed LLM context windows. No analysis on the context length change during the prompt optimization.**
>
> We thank the reviewer for these interesting points. We do not have any context length control in our system. We conducted experiments showing that with context length control, the prompts contain fewer words, and the accuracy drops at the same time, even when instructing LLM to remove distracting details and keep relevant ones.
>
> | Task                      | Word Count w/o Length Control | Word Count w/ Length Control | Acc(%) w/o Length Control | Acc(%) w/ Length Control |
> |--------------------------|-------------------------------|-----------------------------|---------------------------|--------------------------|
> | **College Biology**          | 568                           | 73                          | 96.5                      | 95.14                    |
> | **Machine Learning**         | 424                           | 184                         | 93.8                      | 85.71                    |
> | **College Computer Science** | 906                           | 121                         | 93                        | 85                       |
>
> **6. Duplicated equation indexes in Line 161 and Line 188. Also no equation indexing in Line 194.**
>
> We appreciate the reviewer's detailed observation. We fixed this issue, thanks for the suggestions!

---

### Official Review · Reviewer_pPqD · 2025-10-31

**Soundness:** 3
**Presentation:** 3
**Contribution:** 3
**Rating:** 6
**Confidence:** 3

**Summary:**

This paper introduces SIPDO (Self-Improving Prompts through Data-Augmented Optimization), a closed-loop framework for prompt optimization that links a synthetic data generator with a prompt optimizer. The system iteratively generates challenging synthetic examples tailored to a prompt’s current weaknesses; prompt updates are then performed in response to observed failures. The framework integrates dynamic difficulty adjustment and uses synthetic data as a feedback signal, moving beyond static prompt optimization. Across multiple reasoning benchmarks (e.g., MMLU, BIG-Bench, ProofWriter), SIPDO is shown to outperform existing prompt optimization approaches, demonstrating strong generalization and robustness improvements.

**Strengths:**

SIPDO introduces a closed-loop optimization framework that couples a synthetic data generator with a prompt optimizer through a dual-agent collaboration mechanism. The generator dynamically produces challenging samples targeting the current prompt’s weaknesses, while a progressive difficulty parameter ccc enables a curriculum learning strategy from simple to complex tasks. Ablation results demonstrate that this difficulty-gradient method improves average performance by 17.3%–24.3% compared to one-shot extreme sampling.

**Weaknesses:**

1. Although Table 2 provides a comprehensive overview of prompt optimization baselines, the current comparisons mainly cover works from 2022–2024 and lack the inclusion of more recent 2025 methods. In particular, direct comparisons with the latest closed-loop or iterative prompt optimization approaches are missing. Most existing baselines used in this paper focus on heuristic or search-based prompt engineering rather than a fully integrated feedback loop.
2. While the related work section (pp. 2–3) is thorough, it does not sufficiently engage with progress in EM-like optimization procedures or Bayesian optimization–based feedback mechanisms. This omission weakens the paper’s positioning and makes SIPDO’s core feedback-loop concept appear more novel than it actually is, as similar closed-loop designs have recently emerged.
3. Section 3.1 describes sampling from a synthetic generator regularized via KL divergence to mitigate label imbalance and mode collapse. However, key implementation details—such as how the generator is parameterized, instantiated, and updated, especially for more challenging tasks—remain underexplored and would benefit from further clarification.

**Questions:**

see weeknesses.

---

> ### Author Response · Authors · 2025-11-27
> **Rebuttal by Authors**
>
> We appreciate the reviewer’s thoughtful review and is encouraged by the recognition of our closed-loop optimization framework with difficulty gradient as the core component. We are also glad that the reviewer acknowledged the ablation study for difficulty levels. We have carefully considered your questions and address them individually below.
>
> **1. The current comparisons mainly cover works from 2022–2024 and lack the inclusion of more recent 2025 methods. In particular, direct comparisons with the latest closed-loop or iterative prompt optimization approaches are missing. Most existing baselines used in this paper focus on heuristic or search-based prompt engineering rather than a fully integrated feedback loop.**
>
> We thank the reviewer for this thoughtful suggestion. We agree that we need to conduct more recent works as a comparison, particularly works in iterative prompt optimization. We have conducted additional experiments comparing SIPDO with the integrated feedback-loop works SPO [1] and CriSPO [2] using GPT-4o-mini, where SIPDO still consistently outperforms on most tasks and yields the highest in average, leading by 5.28%. If there are more methods that the reviewer prefer us to compare, please let us know.
>
> | Task   | SPO   | CriSPO | SIPDO |
> |--------|-------|--------|-------|
> | **Causal** | 66.32 | 65.26  | 88 |
> | **Object** | 84.9  | 86.7   | 87.5 |
> | **Geome**  | 47.22 | 76.39 | 73.2 |
> | **Penguins** | 91.33 | 90.67 | 92.1 |
> | **Temporal** | 94.6 | 91.3 | 98 |
> | **Epistemic** | 85.33 | 82 | 85.1 |
> | **Average** | 78.28 | 82.05 | 87.32 |
>
> [1] Xiang, Jinyu and Zhang, Jiayi and Yu, Zhaoyang and Teng, Fengwei and Tu, Jinhao and Liang, Xinbing and Hong, Sirui and Wu, Chenglin and Luo, Yuyu(2025). Self-supervised prompt optimization (No. arXiv:2502.06855) https://arxiv.org/pdf/2502.06855
>
> [2] He, Han and Liu, Qianchu and Xu, Lei and Shivade, Chaitanya and Zhang, Yi and Srinivasan, Sundararajan and Kirchhoff, Katrin(2025). CriSPO: Multi-aspect critique-suggestion-guided automatic prompt optimization for text generation (Proceedings of the AAAI Conference on Artificial Intelligence) https://arxiv.org/pdf/2410.02748
>
> **2. It does not sufficiently engage with progress in EM-like optimization procedures or Bayesian optimization–based feedback mechanisms**
>
> We really appreciate the reviewer’s valuable comments. We have added another paragraph for Related Work:
>
> Adopting LLM as a feedback loop to refine prompts has recently emerged. Self-Refine [1] improves the model by generating feedback based on the previously generated outputs and using that feedback to refine the next output. In Promptbreeder [2], it uses an iterative feedback loop to select better prompts from the original prompts and mutation prompts, while Recursive In-Context Learning for Autonomous Prompt Generation in Large Language Models: A Self-Instructed Approach[3] introduces a framework that refines prompts through iterative loops based on generated outputs. More recently, SPO(Self-Supervised Prompt Optimization) [4] uses iterative feedback to refine prompts by comparing the outputs of the current prompt and its revised version without relying on ground truth, and DLPO [5] optimizes prompts within loops by mimicking a deep learning style where it uses textual loss, gradients, and a feedback process to update prompts. CriSPO [6] also extends iterative feedback loop optimization using critique guides to update prompts with multi-metric. Different from all above, SIPDO adopts the iterative feedback loop with a core component difficulty gradient for synthesized data to improve prompts by challenging current prompts with harder and harder tasks.  If there are more methods that the reviewer prefer us to compare, please let us know.
>
> [1] Madaan, Aman and Tandon, Niket and Gupta, Prakhar and Hallinan, Skyler and Gao, Luyu and Wiegreffe, Sarah and Alon, Uri and Dziri, Nouha and Prabhumoye, Shrimai and Yang, Yiming and others(2023). Self-refine: Iterative refinement with self-feedback (Advances in Neural Information Processing Systems) https://arxiv.org/pdf/2303.17651
>
> [2]  Fernando, Chrisantha and Banarse, Dylan and Michalewski, Henryk and Osindero, Simon and Rockt{\"a}schel, Tim(2023). Promptbreeder: Self-referential self-improvement via prompt evolution (No. arXiv: 2309.16797) https://arxiv.org/pdf/2309.16797
> [3] Yilar, Jonathan and Foster, Olivia and Woods, Benjamin(2024). Recursive in-context learning for autonomous prompt generation in large language models: A self-instructed approach (Authorea Preprints) https://d197for5662m48.cloudfront.net/documents/publicationstatus/223260/preprint_pdf/f4b0aeaa803aa7f86054625f8474e4d6.pdf
>
> [4] Xiang, Jinyu and Zhang, Jiayi and Yu, Zhaoyang and Teng, Fengwei and Tu, Jinhao and Liang, Xinbing and Hong, Sirui and Wu, Chenglin and Luo, Yuyu(2025). Self-supervised prompt optimization (No. arXiv:2502.06855) https://arxiv.org/pdf/2502.06855

---

> > ### Author Response · Authors · 2025-11-27
> >
> > [5] Peng, Dengyun and Zhou, Yuhang and Chen, Qiguang and Liu, Jinhao and Chen, Jingjing and Qin, Libo and Che, Wanxiang(2025). Dlpo: Towards a robust, efficient, and generalizable prompt optimization framework from a deep-learning perspective (No. arXiv:2503.13413) https://arxiv.org/pdf/2503.13413
> >
> > [6] He, Han and Liu, Qianchu and Xu, Lei and Shivade, Chaitanya and Zhang, Yi and Srinivasan, Sundararajan and Kirchhoff, Katrin(2025). CriSPO: Multi-aspect critique-suggestion-guided automatic prompt optimization for text generation (Proceedings of the AAAI Conference on Artificial Intelligence) https://arxiv.org/pdf/2410.02748
> >
> > **3. Key implementation details—such as how the generator is parameterized, instantiated, and updated, especially for more challenging tasks—remain underexplored and would benefit from further clarification.**
> >
> > We appreciate the reviewer for this question. We explicitly instruct the prompt to learn from the abstract and core idea of the question and answer pair (x,y) from true distribution S. Additionally, to prevent generating easy tasks repeatedly, we introduce a difficulty gradient in the data generator with the current difficulty level c_n and the maximum difficulty level n (the same as the number of iterations). Current difficulty level c_n and max difficulty level n inform LLM how much harder the generated sample should be compared to the previously generated data. Moreover, we also feed all previously generated data into the prompt. When generating, we ask LLM to produce data that corresponds to the current difficulty level, given all past generated data with their difficulty levels, and we instruct the LLM never to generate the same data again, but instead to produce more challenging examples as the difficulty increases. We added more details for the data generator in the Appendix. You may also find an illustrative example on page 7 under “Causal Judgment Prompt Template”.

---

> > > ### Comment · Reviewer_pPqD · 2025-11-27
> > >
> > > I have reviewed the rebuttal and thank the authors for their efforts. The authors added experiments on the latest work, provided relevant discussion, and clarified experimental details. The response addresses my concerns, and since the original score was already high, I recommend maintaining the original score.

---

### Official Review · Reviewer_8sdx · 2025-11-03

**Soundness:** 3
**Presentation:** 2
**Contribution:** 2
**Rating:** 4
**Confidence:** 4

**Summary:**

This paper proposes SIPDO (Self-Improving Prompts through Data-Augmented Optimization), a closed-loop prompt optimization framework that directly integrates synthetic data generation into the prompt optimization process. The system consists of two synergistic components: a Synthetic Data Generator that deliberately constructs samples with progressive difficulty to expose weaknesses in the current prompt, and an Auto Prompt Optimizer that iteratively refines the prompt through error analysis, recommendation generation, and targeted refinement. This “generate–test–repair–verify” feedback loop enables prompts to self-evolve without requiring external supervision or newly annotated data.

**Strengths:**

- Integrating synthetic data generation with prompt optimization into a dynamic closed-loop framework goes beyond traditional optimization methods that operate on static datasets.
- Difficulty is monotonically increased through a difficulty parameter and curriculum learning, aligning with principles of human learning.
- The ablation studies are well-designed and effectively validate the contributions of the core components.

**Weaknesses:**

- Each optimization iteration requires multiple LLM invocations, resulting in significantly higher computational cost compared to conventional approaches. Although the three-expert verification mechanism improves data quality, it substantially increases latency.
- Generation quality is directly constrained by the capabilities of the underlying LLM; for instance, GPT-4o-mini performs markedly worse than GPT-4o. The paper does not explore strategies to reduce reliance on powerful base models, limiting applicability in resource-constrained settings.
- Task-specific safeguards (e.g., for geometric SVG generation) demand substantial domain expertise, and the paper lacks a systematic methodology for transferring the framework to new domains, thereby restricting its practical applicability.

**Questions:**

- How does the computational overhead of the framework scale as task complexity increases? The authors should supplement their experiments with an end-to-end analysis that explicitly illustrates the trade-off between optimization time and performance gains.
- For tasks requiring specialized domain knowledge (e.g., medical diagnosis), how can the framework ensure the factual and professional accuracy of generated synthetic samples? Could a general-purpose domain adaptation module be designed to reduce reliance on manually crafted, task-specific safeguards (e.g., the three safeguards for geometric SVG generation)?
- At which stages would human expert intervention be most effective—synthetic data generation, error analysis, or prompt editing—and how might such human-in-the-loop feedback be integrated efficiently?
- As the input distribution continuously evolves over time, how can SIPDO be extended to support continual prompt optimization while mitigating catastrophic forgetting of previously acquired knowledge?
- When encountering a new task category, must the optimization process be restarted from scratch? Is there a mechanism to retain and transfer the general reasoning capabilities already learned by the prompt across tasks?

---

> ### Author Response · Authors · 2025-11-27
> **Rebuttal by Authors**
>
> We thank the reviewer for the positive comments and the thorough review. We appreciate the recognition of the system design with synthetic data and prompt optimization in a closed loop, and also the ablation study that demonstrates the effectiveness of the design of the difficulty level. We also thank the reviewer for their insightful comments and feedback. We believe all the questions and weaknesses can be fully addressed, and we provide a detailed response below.
>
> **1. Each optimization iteration requires multiple LLM invocations, resulting in significantly higher computational cost compared to conventional approaches.**
>
> Thank you for this critical point. We agree that a single optimization iteration of SIPDO is more computationally intensive than that of simpler methods. However, we argue that this higher per-iteration cost enables a far more efficient search for a high-performance prompt, which ultimately leads to a better result with a lower overall cost when compared to conventional approaches.
>
> To put this value clearer, we conduct a direct comparison with PromptAgent, another prevailing prompt optimization method to show their time cost in seconds by GPT-4o and GPT-4o-mini. The result shows that SIPDO runs in much less time while achieving higher accuracy.
>
> | Model        | Method     | Penguins RunTime(s) | Penguins Acc(%) | Epistemic RunTime(s) | Epistemic Acc(%) | Geometric RunTime(s) | Geometric Acc(%) | Temporal RunTime(s) | Temporal Acc(%) | Causal RunTime(s) | Causal Acc(%) | Object RunTime(s) | Object Acc(%) |
> |-------------|------------|-------------------|----------------|---------------------|-----------------|---------------------|-----------------|---------------------|----------------|------------------|----------------|------------------|----------------|
> | GPT-4o   | **PromptAgent** | 18300             | 96.1           | 23675               | 91.6            | 30639               | 83.0            | 40399               | 98.4           | 18232            | 77.8           | 37453            | 88.2           |
> | GPT-4o   | **SIPDO**       | 2392              | 96.4           | 556                 | 86.3            | 1335                | 82.2            | 647                 | 99.3           | 156              | 79.0           | 1094             | 91.1           |
> | GPT-4o-mini | **PromptAgent** | 9398              | 89.8           | 24150               | 86.0            | 46012               | 72.0            | 48561               | 94.6           | 15097            | 84.6           | 41977            | 84.3           |
> | GPT-4o-mini | **SIPDO**     | 2227              | 92.1           | 686                 | 85.1            | 2504                | 73.2            | 147                 | 98.0           | 899              | 88.0           | 512              | 87.5           |
>
> This efficiency gain stems from SIPDO's targeted feedback loop that directly identifies and refines failure points, which allows it to converge on a superior prompt more quickly and reliably, thus avoiding the broader, more costly exploratory processes found in multi-agent optimization systems like PromptAgent.

---

> ### Author Response · Authors · 2025-11-27
>
> **2. How does the computational overhead of the framework scale as task complexity increases? The authors should supplement their experiments with an end-to-end analysis that explicitly illustrates the trade-off between optimization time and performance gains.**
>
> We thank the reviewer for this valuable suggestion. We conducted experiments to show the trade-off between time cost and performance gains made by changing the difficulty level. We compared different difficulty levels in terms of cost and accuracy across 6 tasks on BIG-Bench. We randomly sampled 300 tasks for Object Counting, Epistemic, Temporal, and Causal Judgment, and used the original number of tasks for Geometry and Penguins In a Table.
>
> | Task(By GPT-4o-mini)            | Difficulty | Cost Time(s) | Cost Money($) | Cost Tokens | Accuracy(%) |
> |----------------|------------|--------------|----------------|-------------|-------------|
> | Penguins       | 5          | 217.43       | 0.0130         | 67269       | 83.7        |
> | Penguins       | 10         | 1078.47      | 0.0929         | 527358      | 92.59       |
> | Penguins       | 15         | 2226.68      | 0.2174         | 1250646     | 92.59       |
> | Penguins       | 20         | 19753.99     | 2.6698         | 1669561     | 93.33       |
> | Epistemic      | 5          | 49.07        | 0.0027         | 14297       | 75.33       |
> | Epistemic      | 10         | 686.18       | 0.0420         | 227607      | 76.00       |
> | Epistemic      | 15         | 1471.69      | 0.1149         | 640713      | 80.33       |
> | Epistemic      | 20         | 3279.38      | 0.2583         | 1483605     | 80.00       |
> | Geometric      | 5          | 51.70        | 0.00369        | 19079       | 35.33       |
> | Geometric      | 10         | 690.22       | 0.03237        | 144212      | 49.17       |
> | Geometric      | 15         | 1326.86      | 0.06710        | 296540      | 72.22       |
> | Geometric      | 20         | 2503.59      | 0.20610        | 571178      | 78.06       |
> | Causal Judgment| 5          | 103.80       | 0.0064         | 35013       | 58.42       |
> | Causal Judgment| 10         | 899.33       | 0.0699         | 385346      | 66.66       |
> | Causal Judgment| 15         | 2066.64      | 0.2017         | 1152961     | 79.67       |
> | Causal Judgment| 20         | 443.71       | 0.0388         | 236561      | 67.89       |
> | Temporal       | 5          | 35.40        | 0.0016         | 7650        | 94.00       |
> | Temporal       | 10         | 147.00       | 0.0066         | 26538       | 97.50       |
> | Temporal       | 15         | 2166.98      | 0.1876         | 1039249     | 99.67       |
> | Temporal       | 20         | 1518.23      | 0.1187         | 630972      | 99.64       |
> | Object Counting| 5          | 190.52       | 0.0122         | 60765       | 91.00       |
> | Object Counting| 10         | 1094.08      | 0.0684         | 363795      | 92.00       |
> | Object Counting| 15         | 2592.44      | 0.2069         | 1148406     | 96.00       |
> | Object Counting| 20         | 5078.75      | 0.3968         | 2260081     | 96.00       |

---

> ### Author Response · Authors · 2025-11-27
>
> | Task(By GPT-4o)            | Difficulty | Cost Time(s) | Cost Money($) | Cost Tokens | Accuracy(%) |
> |----------------|------------|--------------|----------------|-------------|-------------|
> | Penguins       | 5          | 58.55        | 0.1089         | 35426       | 91.3        |
> | Penguins       | 10         | 561.05       | 1.21           | 410807      | 93.33       |
> | Penguins       | 15         | 2392.23      | 6.4528         | 2275582     | 96.00       |
> | Penguins       | 20         | 3452.01      | 7.8194         | 2759306     | 96.67       |
> | Epistemic      | 5          | 119.28       | 0.1401         | 42985       | 78.00       |
> | Epistemic      | 10         | 555.86       | 0.6814         | 224480      | 79.00       |
> | Epistemic      | 15         | 1440.52      | 2.0357         | 695678      | 83.67       |
> | Epistemic      | 20         | 2885.80      | 4.0457         | 1418053     | 84.00       |
> | Geometric      | 5          | 106.06       | 0.0955         | 27548       | 50.00       |
> | Geometric      | 10         | 560.95       | 0.4726         | 133780      | 67.67       |
> | Geometric      | 15         | 964.28       | 0.9158         | 268912      | 80.00       |
> | Geometric      | 20         | 1334.71      | 1.4882         | 450575      | 87.78       |
> | Causal Judgment| 5          | 149.56       | 0.1611         | 52383       | 67.95       |
> | Causal Judgment| 10         | 156.22       | 0.2761         | 101715      | 81.60       |
> | Causal Judgment| 15         | 534.45       | 0.6921         | 237519      | 82.42       |
> | Causal Judgment| 20         | 3033.81      | 4.8447         | 1738332     | 79.49       |
> | Temporal       | 5          | 39.27        | 0.0283         | 7813        | 99.67       |
> | Temporal       | 10         | 647.34       | 0.8005         | 267894      | 99.33       |
> | Temporal       | 15         | 1432.08      | 2.1206         | 736935      | 99.33       |
> | Temporal       | 20         | 2235.50      | 3.5084         | 1255135     | 99.66       |
> | Object Counting| 5          | 115.30       | 0.1660         | 49776       | 97.00       |
> | Object Counting| 10         | 512.40       | 0.7296         | 232588      | 98.00       |
> | Object Counting| 15         | 1396.33      | 1.8831         | 624918      | 99.33       |
> | Object Counting| 20         | 2848.81      | 4.3167         | 1459739     | 99.33       |
>
> From the experiments, we observe performance consistently improves as the difficulty level (and thus, the computational cost) increases. Crucially, we also observe that the performance gains diminish as the difficulty level approaches 20.
>
> **3. Although the three-expert verification mechanism improves data quality, it substantially increases latency.**
>
> Thank you for mentioning this. In an iterative system like SIPDO, a single low-quality example could corrupt the entire optimization trajectory. The voter mechanism acts as a gatekeeper to prevent this compounding error. To make this value clearer, we conduct experiments to show the trade-off between accuracy, latency, and cost when removing three expert voters in 3 MMLU tasks (college-level biology, computer science, and machine learning).
>
> | Task                     | Time(s) Remove Voters | Time(s) With Voters | Money($) Remove Voters | Money($) With Voters | Perf(%) Remove Voters | Perf(%) With Voters |
> |-------------------------|----------------|--------------|-----------------|----------------|----------------|--------------|
> | **College Biology**         | 763            | 912          | 0.0167          | 0.0221         | 95.14          | 96.5         |
> | **Machine Learning**        | 375            | 680          | 0.0096          | 0.7974         | 76.79          | 93.8         |
> | **College Computer Science**| 432            | 1311         | 0.0104          | 1.808          | 88.00          | 93.0         |
>
> We agree that the latency is the explicit trade-off for the data quality provided by the three experts. However, we kindly argue that this upfront latency is a strategic investment to prevent much larger, wasted computational costs in the later stages of the refinement loop.

---

> > ### Author Response · Authors · 2025-11-27
> >
> > **4. Generation quality is directly constrained by the capabilities of the underlying LLM; for instance, GPT-4o-mini performs markedly worse than GPT-4o. The paper does not explore strategies to reduce reliance on powerful base models, limiting applicability in resource-constrained settings.**
> >
> > This is a crucial point, and our method directly provides a strategy for this exact challenge. We can trade computational budgets for model capability. By increasing the optimization effort (i.e., a higher difficulty level), our method can empower a weaker model like GPT-4o-mini to achieve performance comparable to a much stronger model like GPT-4o.
> >
> > Our results confirm this. For instance, in the Causal Judgement task, GPT-4o-mini with a higher difficulty level (15) achieves 79.67% accuracy, nearly matching GPT-4o at a lower difficulty level (10) which scored 81.6%. The table below shows this is a consistent pattern.
> >
> > | Task(By GPT-4o-mini)            | Difficulty | Cost Time(s) | Cost Money($) | Cost Tokens | Accuracy(%) |
> > |----------------|------------|--------------|----------------|-------------|-------------|
> > | Penguins       | 5          | 217.43       | 0.0130         | 67269       | 83.7        |
> > | Penguins       | 10         | 1078.47      | 0.0929         | 527358      | 92.59       |
> > | Penguins       | 15         | 2226.68      | 0.2174         | 1250646     | 92.59       |
> > | Penguins       | 20         | 19753.99     | 2.6698         | 1669561     | 93.33       |
> > | Epistemic      | 5          | 49.07        | 0.0027         | 14297       | 75.33       |
> > | Epistemic      | 10         | 686.18       | 0.0420         | 227607      | 76.00       |
> > | Epistemic      | 15         | 1471.69      | 0.1149         | 640713      | 80.33       |
> > | Epistemic      | 20         | 3279.38      | 0.2583         | 1483605     | 80.00       |
> > | Geometric      | 5          | 51.70        | 0.00369        | 19079       | 35.33       |
> > | Geometric      | 10         | 690.22       | 0.03237        | 144212      | 49.17       |
> > | Geometric      | 15         | 1326.86      | 0.06710        | 296540      | 72.22       |
> > | Geometric      | 20         | 2503.59      | 0.20610        | 571178      | 78.06       |
> > | Causal Judgment| 5          | 103.80       | 0.0064         | 35013       | 58.42       |
> > | Causal Judgment| 10         | 899.33       | 0.0699         | 385346      | 66.66       |
> > | Causal Judgment| 15         | 2066.64      | 0.2017         | 1152961     | 79.67       |
> > | Causal Judgment| 20         | 443.71       | 0.0388         | 236561      | 67.89       |
> > | Temporal       | 5          | 35.40        | 0.0016         | 7650        | 94.00       |
> > | Temporal       | 10         | 147.00       | 0.0066         | 26538       | 97.50       |
> > | Temporal       | 15         | 2166.98      | 0.1876         | 1039249     | 99.67       |
> > | Temporal       | 20         | 1518.23      | 0.1187         | 630972      | 99.64       |
> > | Object Counting| 5          | 190.52       | 0.0122         | 60765       | 91.00       |
> > | Object Counting| 10         | 1094.08      | 0.0684         | 363795      | 92.00       |
> > | Object Counting| 15         | 2592.44      | 0.2069         | 1148406     | 96.00       |
> > | Object Counting| 20         | 5078.75      | 0.3968         | 2260081     | 96.00       |

---

> ### Author Response · Authors · 2025-11-27
>
> | Task(By GPT-4o)            | Difficulty | Cost Time(s) | Cost Money($) | Cost Tokens | Accuracy(%) |
> |----------------|------------|--------------|----------------|-------------|-------------|
> | Penguins       | 5          | 58.55        | 0.1089         | 35426       | 91.3        |
> | Penguins       | 10         | 561.05       | 1.21           | 410807      | 93.33       |
> | Penguins       | 15         | 2392.23      | 6.4528         | 2275582     | 96.00       |
> | Penguins       | 20         | 3452.01      | 7.8194         | 2759306     | 96.67       |
> | Epistemic      | 5          | 119.28       | 0.1401         | 42985       | 78.00       |
> | Epistemic      | 10         | 555.86       | 0.6814         | 224480      | 79.00       |
> | Epistemic      | 15         | 1440.52      | 2.0357         | 695678      | 83.67       |
> | Epistemic      | 20         | 2885.80      | 4.0457         | 1418053     | 84.00       |
> | Geometric      | 5          | 106.06       | 0.0955         | 27548       | 50.00       |
> | Geometric      | 10         | 560.95       | 0.4726         | 133780      | 67.67       |
> | Geometric      | 15         | 964.28       | 0.9158         | 268912      | 80.00       |
> | Geometric      | 20         | 1334.71      | 1.4882         | 450575      | 87.78       |
> | Causal Judgment| 5          | 149.56       | 0.1611         | 52383       | 67.95       |
> | Causal Judgment| 10         | 156.22       | 0.2761         | 101715      | 81.60       |
> | Causal Judgment| 15         | 534.45       | 0.6921         | 237519      | 82.42       |
> | Causal Judgment| 20         | 3033.81      | 4.8447         | 1738332     | 79.49       |
> | Temporal       | 5          | 39.27        | 0.0283         | 7813        | 99.67       |
> | Temporal       | 10         | 647.34       | 0.8005         | 267894      | 99.33       |
> | Temporal       | 15         | 1432.08      | 2.1206         | 736935      | 99.33       |
> | Temporal       | 20         | 2235.50      | 3.5084         | 1255135     | 99.66       |
> | Object Counting| 5          | 115.30       | 0.1660         | 49776       | 97.00       |
> | Object Counting| 10         | 512.40       | 0.7296         | 232588      | 98.00       |
> | Object Counting| 15         | 1396.33      | 1.8831         | 624918      | 99.33       |
> | Object Counting| 20         | 2848.81      | 4.3167         | 1459739     | 99.33       |
>
> **5. For tasks requiring specialized domain knowledge (e.g., medical diagnosis), how can the framework ensure the factual and professional accuracy of generated synthetic samples?**
>
> We thank the reviewer for this interesting point. Our framework addresses this directly via the modularity of the voter mechanism.
>
> The voters are not fixed; they are pluggable expert slots. This provides two clear paths to ensure domain-specific accuracy:
>
> * Expert Prompting: Voters can be instructed with expert personas (e.g., "act as a medical expert") to leverage the base model's latent knowledge for validation.
>
> * External Tool Integration: A vote' can be an API call to a specialized tool, such as a medical rule-based engine [1] or a fact-checking database [2].
>
> This flexible design makes it inherently extensible to high-stakes domains.
>
> [1] Ilaty, Arshia and Shirazi, Hossein and Homayouni, Hajar(2025). SynLLM: A Comparative Analysis of Large Language Models for Medical Tabular Synthetic Data Generation via Prompt Engineering (No. arXiv:2508.08529) https://arxiv.org/pdf/2508.08529
>
> [2] Chung, Philip and Swaminathan, Akshay and Goodell, Alex J and Kim, Yeasul and Reincke, S Momsen and Han, Lichy and Deverett, Ben and Sadeghi, Mohammad Amin and Ariss, Abdel-Badih and Ghanem, Marc and others(2025). Verifact: Verifying facts in llm-generated clinical text with electronic health records (No. arXiv:2501.16672) https://arxiv.org/pdf/2501.16672
>
> **6. Could a general-purpose domain adaptation module be designed to reduce reliance on manually crafted, task-specific safeguards (e.g., the three safeguards for geometric SVG generation)**
>
> We thank the reviewer for this thoughtful question. Inspired by this suggestion, we conducted experiments with an adaptation module. We provide an example of the prompt produced by the adaptation module below for Geometric task below, the final accuracy achieved with the adaptation module is 57%(GPT-4o-mini), surpassing two baselines from the paper.

---

> ### Author Response · Authors · 2025-11-27
>
> Specifically, in each iteration, the adaptation module draws 20 random samples from the true data $S$ to serve as few-shot examples. The module then analyzes these samples to extract the structure and content of the true data $S$. This process is flexible for each difficulty level: the module rewrites the prompt for constraints designs increasingly difficult data generation guidelines for that specific difficulty level automatically. For instance, the number of SVG commands or the change degree of geometric shapes. This entire module crafts a data generator prompt based on the current difficulty level and true data $S$ without any human intervention. For each difficulty level, the adaptation module updates the data generator prompt to align with the most current difficulty level.
>
> ==== Difficulty 9====
>
> You are an SVG path planner that leverages dataset-derived heuristics instead of manual
> instructions. Generate a brand-new SVG path for a 'hexagon'.
>
> Shape heuristics (auto-mined):
> - polygonal contour; minimal arc usage; often open paths; avg command count ≈ 8.5.
>
> Reference fragments (auto-selected):
> - d="M 59.43,52.76 L 75.49,27.45 L 54.92,4.40 M 54.92,4.40 L 23.70,7.77 L 15.15,42.15 L 34.51,57.44 L 59.43,52.76"
> - d="M 50.91,18.41 L 57.39,58.34 L 25.82,45.12 L 33.11,31.36 L 26.90,27.04 L 29.87,20.84 M 29.87,20.84 L 50.91,18.41"
> - d="M 29.25,88.14 L 74.91,74.60 M 74.91,74.60 L 84.13,44.02 M 84.13,44.02 L 75.16,14.24 L 45.19,61.73 M 45.19,61.73 L 25.89,86.05 L 29.25,88.14"
>
> Difficulty constraints (level 9):
> - Complexity tier: structured (level 9/20).
> - Use ≥ 19 drawing commands with 2-decimal precision.
> - Keep coordinates in the 0-100 canvas and jitter edges by ±3.2 units to avoid symmetry.
> - Include at least one 'A' arc blended with straight lines.
> - Close the outer contour with 'Z' unless drawing a line.
> - Coordinates must not be whole numbers; use decimal notation for every literal.
> - Introduce variations in edge length/angle so each attempt feels harder than the previous level.
> - Keep the answer strictly in the form: This SVG path element <path d="..."/> draws a
> - Do not echo explanations or extra text.
>
> This result suggests that our framework is effective even with machine-generated safeguards, which reduces the need for manual crafting and enhances the framework's general-purpose applicability.
>
> **7. At which stages would human expert intervention be most effective—synthetic data generation, error analysis, or prompt editing—and how might such human-in-the-loop feedback be integrated efficiently?**
>
> We appreciate the reviewer for pointing this out. We provide experiments below by removing human guidance from each stage independently. Our findings allow us to rank the impact of human intervention:
>
> 1. Prompt Editing (Most Critical): When removing human guidance on prompt editing, the accuracy drops significantly. This indicates prompt editing stage benefits the most from human intervention by providing general principles in the prompt.
> 2. Error Analysis (Second Most Critical): Without human guidance on the kind of general issues to pay attention to, accuracy also declines and time cost increases at the same time.
> 3. Synthetic Data Generation (Least Critical): Removing human predefined task type during the data generation stage has the least effect on performance. By providing few-shot examples from the true dataset $S$, LLM can learn efficiently and generate valid data.
>
> | Task                      | Acc w/o Generator | Acc w/o Error Analysis | Acc w/o Revisor | Time w/o Generator (s) | Time w/o Error Analysis (s) | Time w/o Revisor (s) |
> |--------------------------|------------------|-----------------------|-----------------|------------------------|-----------------------------|----------------------|
> | **College Biology**          | 95.83            | 95.83                 | 95.14           | 331                    | 677                         | 314                  |
> | **Machine Learning**         | 90.18            | 84.82                 | 77.68           | 767                    | 1622                        | 353                  |
> | **College Computer Science** | 94.00            | 90.00                 | 89.00           | 2405                   | 2852                        | 2861                 |

---

> ### Author Response · Authors · 2025-11-27
>
> Based on this ranking, an efficient integration would be a targeted intervention workflow. Instead of constant supervision, the system would run autonomously and flag critical events for expert review, such as: 1) recurring error patterns it cannot resolve and 2) performance stagnation requiring a creative leap in prompt editing. This concentrates expert effort on the most critical stages for maximum efficiency.
>
> **8. As the input distribution continuously evolves over time, how can SIPDO be extended to support continual prompt optimization while mitigating catastrophic forgetting of previously acquired knowledge?**
>
> We thank the reviewer for this critical question. Our SIPDO has an inherent mechanism to address this, which is analogous to the experience replay strategy used in continual learning to mitigate catastrophic forgetting.
>
> The set of all previously generated and validated synthetic data acts as a replay buffer. When a prompt is revised to handle a new failure case (adapting to a new distribution), our method enforces a strict constraint: the revised prompt must still pass validation against this entire historical data buffer. This process explicitly prevents the optimizer from overfitting to the new dataset at the expense of forgetting old ones.
>
> | Task                      | Accuracy w/o Revising on Generated Cases | Accuracy with Revising on Generated Cases |
> |--------------------------|------------------------------------------|-------------------------------------------|
> | **College Biology**          | 93.75                                    | 96.5                                      |
> | **Machine Learning**         | 82.14                                    | 93.8                                      |
> | **College Computer Science** | 74                                       | 93                                        |
>
> **9. When encountering a new task category, must the optimization process be restarted from scratch? Is there a mechanism to retain and transfer the general reasoning capabilities already learned by the prompt across tasks?**
>
> This is an excellent question about the transferability of learned prompt structures across tasks. To investigate this, we designed an experiment to test if previously optimized prompts can serve as a warm start for new tasks, even across different domains.
>
> During the prompt refinement stage for a new task (e.g., Computer Science from MMLU), we provided the LLM with previously optimized prompts from other domains (e.g., Biology, ML) as in-context examples of high-quality reasoning structures.
>
> The meta-prompt did not ask the LLM to apply biological knowledge. Instead, it instructed the LLM to "analyze the structural and strategic patterns from these expert-level examples and adapt similar strategies to solve the new computer science problem.
>
> | Task                      | Acc(%) From Meta-prompt | Acc(%) From Scratch | Time(s) From Meta-prompt | Time(s) From Scratch |
> |--------------------------|-----------------------|--------------------|--------------------------|---------------------|
> | **College Biology**          | 95.83                 | 96.50              | 1106                     | 912                 |
> | **Machine Learning**         | 91.07                 | 93.80              | 1723                     | 680                 |
> | **College Computer Science** | 91.00                 | 93.00              | 1606                     | 1311                |

---

### Author Response · Authors · 2025-12-01
**Author Summary**

For reviewer **8sdx**, we addressed all concerns with experiments. We demonstrated SIPDO’s efficiency over other methods and provided a detailed cost-performance trade-off analysis for the difficulty gradient. We also showed the effectiveness of synthetic data generation and the adaptation module as well as domain transfer capability, quantified where human intervention is most impactful, and showed SIPDO’s inherent mitigation of catastrophic forgetting through continual validation on historical data. For reviewer **pPqD**, we added the latest closed-loop prompt optimization baselines (SPO, CriSPO) into comparisons while providing more references on the related work with recent feedback-loop and EM-style optimization papers. We also further clarified the implementation of the data generator with added details and examples. For reviewer **E8bU**, we provided variance analysis as a quantitative measure of robustness under distribution shift, analyzed the cost-performance trade-off for different difficulty levels, conducted context-length analysis, and made clarifications and revisions on supervision and writing issues.

**As a result, reviewer pPqD has already confirmed that we addressed his/her concerns either by providing experiments or references. Reviewer E8bU also indicated a willingness to change the score based on our response.**

---

### Meta-Review · Area_Chair_KCvo · 2026-01-05

**Summary:**

The main reviewer concerns centered on three points: (i) computational cost and scalability of the closed-loop optimization as difficulty increases, (ii) novelty and positioning relative to recent closed-loop or EM-style prompt optimization methods, and (iii) clarity and rigor of implementation details, including supervision assumptions, difficulty scheduling, robustness under distribution shift, and potential information leakage or context-length issues.

**Reviewer Concerns:**

1. Reviewer 8sdx raised concerns about high per-iteration cost, reliance on strong base models, domain-specific safeguards, continual learning, and transfer. These were largely addressed by extensive new cost–performance analyses, difficulty–accuracy trade-offs, experiments showing weaker models catching up with higher difficulty, modular expert voters, adaptation modules, replay-style mitigation of forgetting, and warm-start transfer experiments. Overall, the substantive concerns appear resolved.

2. Reviewer pPqD questioned missing comparisons with 2025 closed-loop methods and insufficient engagement with EM-like or Bayesian optimization literature, as well as unclear generator implementation. The rebuttal added direct comparisons with SPO and CriSPO, expanded related work, and clarified generator parameterization and difficulty gradients. The reviewer explicitly confirmed that concerns were addressed and recommended keeping the original score.

3. Reviewer E8bU questioned claims of no supervision, lack of quantitative evidence under distribution shift, definition and role of difficulty level, possible information leakage in the recommendation step, and lack of context-length analysis. The authors revised claims to “minimal supervision,” added variance-based robustness analysis, clarified difficulty scheduling, fixed presentation issues, and provided empirical analysis on context-length control. These responses directly addressed the raised issues, and the reviewer had indicated willingness to revise the score.

**Reviewer Scores:**

Reviewer 8sdx: Likely remains around the original marginally negative assessment, though several major weaknesses were mitigated by new experiments and analyses.

Reviewer pPqD: Explicitly stated the rebuttal addressed concerns and recommended maintaining the original, slightly positive score.

Reviewer E8bU: Given that all listed technical and clarity concerns were directly addressed with new experiments and corrections, a modest upward adjustment from the original marginally negative stance is plausible.

---

### Decision · Program_Chairs · 2026-01-26

Accept (Poster)